# SIKA-GP: Accelerating Gaussian Process Inference with Sparse Inducing Kernel Approximations for Bayesian Deep Learning

Wenyuan Zhao [1]    Rui Tuo [2]    Chao Tian [1]

## Abstract

Gaussian processes (GPs) provide a principled Bayesian framework for uncertainty estimation, but their computational complexity severely limits scalability to large datasets. We propose SIKA-GP, which accelerates GP inference using sparse inducing kernel approximations based on a dyadic ordered template basis, incurring only $O(\log M)$ complexity dependence on the number of inducing points. Our approach constructs compact and expressive kernel representations from sparsely activated bases, enabling efficient tensorized GPU computation and seamless integration with modern large-scale models. SIKA-GP can be naturally embedded into Bayesian neural networks (BNNs) with sparse activations, yielding significant speedups in both training and inference without sacrificing predictive performance. The method naturally extends to deep feature learning, addressing the scalability challenges introduced by deep architectures and high-dimensional feature representations. Empirical results on vision and transformer-based language benchmarks demonstrate that our approach consistently delivers fast and accurate GP models, providing a principled path toward scalable kernel learning.

## 1. Introduction

Gaussian processes (GPs) (Rasmussen & Williams, 2006) provide a powerful Bayesian nonparametric framework for supervised learning, offering calibrated uncertainty estimates and strong generalization performance. However, the cubic complexity $O(N^3)$ in the number of training points $N$, due to the kernel matrix inverse, renders the exact inference computationally infeasible for large-scale datasets.

To address this challenge, *sparse* GPs (SGPs) approximate the full covariance using a smaller set of inducing points (Quinonero-Candela & Rasmussen, 2005; Snelson & Ghahramani, 2006). Variational formulations (Titsias, 2009; Hensman et al., 2013) and structured kernel methods (Wilson & Nickisch, 2015; Wilson et al., 2020) have pushed the scalability of GPs to larger datasets. Ding et al. (2024) and Zhao et al. (2025) convert SGPs into Bayesian neural networks (BNNs) through induced kernel approximation, thus harnessing the advantages of parallel computing found in NNs. However, these approaches still face difficulties when applied to very high-dimensional feature spaces or when many inducing points are required, which limits both computational efficiency and predictive accuracy.

In this work, we propose a new algorithm to accelerate GP inference through *sparse inducing kernel approximation* (SIKA-GP). Our approach takes advantage of the sparse structure of the dyadic kernel basis, producing significant improvements in both training and inference[1] speed while preserving predictive accuracy. Importantly, the method naturally extends to applications in Bayesian deep learning (BDL), such as DGPs and DKL, where scalability is a central obstacle. Furthermore, we show that our approximation integrates seamlessly into sparsely activated Bayesian neural network (BNN) architectures, enabling hybrid deterministic-probabilistic models that combine uncertainty quantification with deep feature learning. We summarize our contributions as follows.

- We propose SIKA-GP designed on a set of compactly supported basis functions with closed-form expressions, providing accelerated GP inference.
- We extend SIKA-GP to deep feature learning, where scalability challenges are particularly acute due to high dimensionality and large-scale datasets.
- We demonstrate through theoretical and experimental analysis that our approach achieves significant speedups over existing dense and sparse GP baselines, while maintaining predictive accuracy.

Finally, we release the data and code for SIKA-GP to encourage further studies of efficient inference and un-

---

[1]Department of Electrical and Computer Engineering, Texas A&M University, College Station, US [2]Department of Industrial and Systems Engineering, Texas A&M University, College Station, US. Correspondence to: Wenyuan Zhao <wyzhao@tamu.edu>.

*Proceedings of the 43$^{rd}$ International Conference on Machine Learning*, Seoul, South Korea. PMLR 306, 2026. Copyright 2026 by the author(s).

---

[1]We use "test" and "inference" interchangeably in this work.

certainty estimation in BDL: https://github.com/warrenzha/sika-gp.

## 2. Preliminaries

**Gaussian Processes.** A GP $f(\mathbf{x})$ is fully characterized by its mean function $\mu(\mathbf{x})$ and its covariance function (kernel) $K(\mathbf{x}, \mathbf{x}')$. Given a dataset $\mathcal{D} = \{\mathbf{X}, \mathbf{y}\}$, $\mathbf{X} \in \mathbb{R}^{N \times D}$ are $N$ training points, of which $\mathbf{x}_i \in \mathbb{R}^D$ is the $D$-dimensional feature, and $\mathbf{y} = [y_1, \dots, y_N]^\top$ is the corresponding noisy observation where $y_i \in \mathbb{R}$. We assume that $\mu(\mathbf{x})$ is a constant (usually assigned to zero). In exact GPs where the likelihood $P(\mathbf{y}|f(\mathbf{X}))$ is also Gaussian, the predictive posterior $P(\mathbf{f}^*|\mathbf{X}^*, \mathbf{y}, \mathbf{X})$ at $N^*$ test points $\mathbf{X}^* \in \mathbb{R}^{N^* \times D}$ is also Gaussian, which can be written in a closed form involving the inverse of the covariance matrix at the observation points. The computational complexity of exact GPs scales cubically in the number of training points $O(N^3)$, which is impractical for large datasets without approximations. For non-regression tasks or deep Gaussian processes, posteriors are non-Gaussian and usually require additional approximation or other treatments. We refer the reader to Williams & Rasmussen (2006) for more details.

**DGP and DKL.** To address the limited expressiveness of GPs in complex, nonstationary, or hierarchical data, several extensions of GPs have been developed to integrate them into the Bayesian deep learning landscape. DGPs (Damianou & Lawrence, 2013) extend standard GPs by stacking GPs in multiple layers, where the output of one layer serves as input to the next:

$$f(\mathbf{x}) = f^{(H)} \circ f^{(H-1)} \circ \cdots \circ f^{(1)}(\mathbf{x}). \quad (1)$$

DKL (Wilson et al., 2016), on the other hand, maps raw inputs into a feature space using deep neural networks (DNNs), followed by a single layer of GP:

$$f(\mathbf{x}) \sim \mathcal{GP}\left(\mathbf{0}, K(\text{NN}(\mathbf{x}), \text{NN}(\mathbf{x}'))\right). \quad (2)$$

**Additive GP.** To address the curse of dimensionality, an additive approximation is commonly used to decompose GPs with high-dimensional features into first-order additive GPs (Duvenaud et al., 2011). For input $\mathbf{x} \in \mathbb{R}^D$, i.e., $D$-dimensional features, the additive GP can be expressed by

$$f(\mathbf{x}) = \sum_{d=1}^{D} f_d(x_d), \quad K(\mathbf{x}, \mathbf{x}') = \sum_{d=1}^{D} K_d(x_d, x_d'), \quad (3)$$

where $f_d(x_d)$ denotes a *base* GP.

**Connecting GP to BNN.** To reduce computational burden of exact GPs, inducing point methods approximate a GP $f(\mathbf{x})$ using $M$ inducing points $\mathbf{U} = \{\mathbf{u}_i \in \mathbb{R}^D\}_{i=1}^M$:

$$f(\mathbf{x}) := K(\mathbf{x}, \mathbf{U})[K(\mathbf{U}, \mathbf{U})]^{-1} f(\mathbf{U}).$$

Suppose $K(\mathbf{U}, \mathbf{U}) = \mathbf{P}_\mathbf{U} \mathbf{P}_\mathbf{U}^\top$, then this factorization allows us to rewrite $f(\mathbf{x})$ as a finite-width BNN

$$\begin{aligned} f(\mathbf{x}) &= K(\mathbf{x}, \mathbf{U}) \left[\mathbf{P}_\mathbf{U} \mathbf{P}_\mathbf{U}^\top\right]^{-1} f(\mathbf{U}) \\ &= \left[K(\mathbf{x}, \mathbf{U}) \mathbf{P}_\mathbf{U}^{-\top}\right] \left[\mathbf{P}_\mathbf{U}^{-1} f(\mathbf{U})\right] \\ &= \phi(\mathbf{x}) \cdot \mathbf{w}, \end{aligned} \quad (4)$$

where $\phi(\mathbf{x}) := K(\mathbf{x}, \mathbf{U}) \mathbf{P}_\mathbf{U}^{-\top}$ is a set of non-linear activation functions (for the fixed kernel and $\mathbf{U}$), and $\mathbf{w} := \mathbf{P}_\mathbf{U}^{-1} f(\mathbf{U}) \sim \mathcal{N}(\mathbf{0}, \mathbf{I})$ are i.i.d. normal random weights. This BNN representation enables efficient training via parallel computing and backpropagation (Wen et al., 2018). It is worth mentioning that $\mathbf{P}_\mathbf{U}$ is *not* unique since $\mathbf{P}_\mathbf{U} \mathbf{P}_\mathbf{U}^\top = [\mathbf{P}_\mathbf{U} \mathbf{O}][\mathbf{P}_\mathbf{U} \mathbf{O}]^\top$ for any orthogonal matrix $\mathbf{O}$, and in turn, the BNN representation in (4) is not unique.

## 3. Related Work

**Inducing point GPs.** To overcome the $O(N^3)$ cost of exact GPs (Rasmussen & Williams, 2006), a large body of work introduces a sparse variational structure through inducing points (Quinonero-Candela & Rasmussen, 2005; Snelson & Ghahramani, 2006; Titsias, 2009), which reduces the inference complexity to $O(NM^2)$ where $M$ is the number of inducing points. Stochastic variational GP (SVGP) enables mini-batch training for large $N$ and classification (Hensman et al., 2013; 2015). Our method shares the goal of reducing complexity but differs in replacing learnable inducing locations with SIKA on a dyadic grid, producing a level of sparsity beyond learning the inducing points alone.

**Structured kernels and interpolation.** Exploiting kernel structure can lead to major computational speedups in GPs: Toeplitz/Kronecker methods (Saatçi, 2012; Wilson et al., 2014), structured kernel interpolation (SKI/KISS-GP) (Wilson & Nickisch, 2015; Pleiss et al., 2018), and exact/near–exact GPU solvers for very large datasets (Gardner et al., 2018; Wilson et al., 2020; Wang et al., 2019). Compactly supported feature functions (Chen et al., 2022; Ding et al., 2024) can lead to sparse matrices in learning with GPs and DGPs.

**GPs and Bayesian deep learning.** DGPs compose multiple GP layers for hierarchical representations (Damianou & Lawrence, 2013), but inference is challenging (Salimbeni & Deisenroth, 2017; Bui et al., 2016; Cutajar et al., 2017). Our approximation is integrated into each separate GP layer, reducing the cost per–layer so that deep architectures become practical. For DKL, scalable implementations on GPUs (Gardner et al., 2018; Charlier et al., 2021) and interpolated covariances (Pleiss et al., 2018; Zhao et al., 2025) integrate DKL with BDL. We show that replacing dense/interpolated covariances with the proposed SIKA yields further training and inference gains, and we demonstrate its compatibility with sparsely–activated BNN backbones.

# 4. SIKA-GP: GP with Sparse Inducing Kernel Approximations

Our goal is to design a computationally efficient framework that can substantially accelerate GPs in both training and inference. In particular, the underlying structured sparsity of GPs can be equivalently represented as *sparsely activated* BNNs in (4), where $\phi(x)$ is the structured sparse activation for each $x$. We refer to this method as GP with sparse inducing kernel approximations (SIKA-GP).

## 4.1. SIKA-GP with Laplace correlation

The main idea of the construction here originates from the sparse representation of Gauss-Markov kernels (Ding et al., 2020; 2024). Here we focus on the Laplace kernel $K(x,y) = \exp(-\theta|x-y|)$, which is Markovian and stationary. Throughout the paper, $L$ denotes the *dyadic level* and $M = 2^L + 1$ denotes the corresponding number of inducing points. We assume that the input $x$ is in $[0,1]^2$ and $M = 2^L + 1$ evenly spaced inducing points are used, given by

$$\mathbf{U} := \{0 \cdot 2^{-L}, 1 \cdot 2^{-L}, \ldots, 2^L \cdot 2^{-L}\}. \quad (5)$$

This framework allows for a BNN representation equipped with explicitly structured and closed-form sparse activation functions, as shown in Theorem 4.1.

**Theorem 4.1.** *Given a Laplace kernel defined by $K(x,y) = \exp(-\theta|x-y|)$ and inducing points $\mathbf{U}$ in (5), there is a sparsely activated BNN in terms of Equation (4) with a set of basis functions $\phi(x) = \{\psi_{lm} : [0,1] \to \mathbb{R}\}$, given by*

$$\psi_{01}(x) := \frac{\exp\{-\theta x\} + \exp\{-\theta(1-x)\}}{\sqrt{2(1 + \exp\{-\theta\})}},$$

$$\psi_{02}(x) := \frac{\exp\{-\theta x\} - \exp\{-\theta(1-x)\}}{\sqrt{2(1 - \exp\{-\theta\})}},$$

*and*

$$\psi_{lm}(x) := \begin{cases} \sqrt{\dfrac{2}{\sinh(2^{1-l}\theta)}} \sinh(\theta(x - (m-1)2^{-l})), \\ \qquad \text{if } (m-1)2^{-l} \le x \le m2^{-l}, \\[2mm] \sqrt{\dfrac{2}{\sinh(2^{1-l}\theta)}} \sinh(\theta((m+1)2^{-l} - x)), \\ \qquad \text{if } m2^{-l} \le x \le (m+1)2^{-l}, \\[2mm] 0, \quad \text{if } x \notin [(m-1)2^{-l}, (m+1)2^{-l}], \end{cases}$$

*for $l = 1, 2, \ldots, L$, and $m = 1, 3, \ldots, 2^l - 1$.*

Theorem 4.1 implies that $\phi(x) = \{\psi_{lm}(x)\} \in \mathbb{R}^{1 \times M}$ is a sparse vector because each $\psi_{lm}$ is supported on a local

---

$^2$For arbitrary input $x \in \mathbb{R}$, we can use a scaling function to normalize $x$ into $[0, 1]$, which is a common choice in SKI.

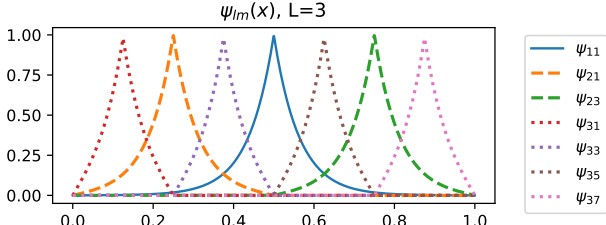

*Figure 1.* An example of basis functions ($L = 3$).

region $[(m-1)2^{-l}, (m+1)2^{-l}]$ whenever $l \ge 1$. An example of $\psi_{lm}(x)$ with $L = 3$ is illustrated in Figure 1.

*Remark* 4.2 (Laplace kernel). The Laplace kernel is a more constrained choice that lacks certain features offered by other, more general kernel classes, for example, the ability to control smoothness. However, there is a significant gain on the $O(\log M)$ scalability because of the Markov property of the Laplace kernel, while other kernels do not. The expressiveness of the Laplace kernel can be enhanced by incorporating modern deep learning, such as DGP and DKL.

*Remark* 4.3 (Inducing grids). Inducing points $\mathbf{U}$ are located uniformly on a predetermined grid rather than being learned adaptively. SIKA-GP allows us to use a much denser grid without sacrificing efficiency: although $M$ basis weights are learned, only $O(\log M)$ basis functions are activated per forward pass. The fixed inducing grid avoids repeated kernel matrix inversion, yielding more stable and efficient optimization with analytic basis activation.

*Remark* 4.4 (Extension to deep architecture). To enhance expressive power and nonstationary behaviors, SIKA-GP can be naturally extended to deep feature learning by stacking multiple layers of SIKA-GP as shown in (1). This hierarchical representation allows us to rewrite a SIKA-DGP as a BNN with additive SIKA-GP blocks.

## 4.2. BNNs with additive SIKA-GP blocks

Given the input $\mathbf{x} \in \mathbb{R}^D$, the output of the additive SIKA-GP blocks is

$$f(\mathbf{x}) = \sum_{i=1}^{D} \phi(x_i)\mathbf{w}_i + b = \Phi(\mathbf{x})\mathbf{W} + b, \quad (6)$$

where $\Phi(\mathbf{x})$ contains a total of $DM$ basis functions when $M$ inducing points are chosen for each base GP, and $\mathbf{W}$ consists of $DM$-dimensional random weights associated with the corresponding basis functions. We place a random bias $b$ to make the structure consistent with vanilla BNN representations.

*Remark* 4.5. Compared with Equation (4) with generic kernels, SIKA-GP leads to significant computation reduction. Since the basis functions $\{\psi_{lm}(x)\}$ have local supports, the activation function has a maximum $(L + 2)$ active units,

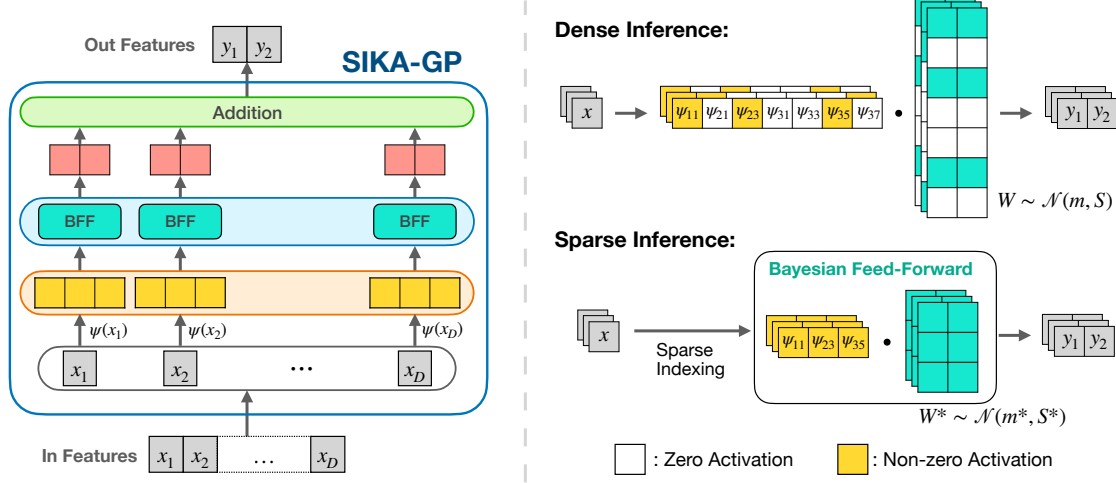

*Figure 2.* SIKA-GP replaces dense GP inference with sparse inducing kernel approximations. A sparse feature mapping selects a small subset of activated basis functions $\psi(x)$ per input, which are processed by Bayesian feed-forward layers and aggregated additively, enabling efficient inducing kernel inference with near-linear complexity.

and the effective random weights $\mathbf{W}^*$ are only $D(L + 2)$-dimensional, i.e., $(L + 2)$-dimensional for each base GP. For a fixed dyadic inducing set, the sparse basis is an orthonormal reparameterization of the same inducing-kernel approximation rather than an additional approximation on top of it; the computational gain comes from evaluating this finite-dimensional span in sparse coordinates. Figure 2 illustrates the architecture of SIKA-GP and the sparsification effect.

### 4.3. Variational inference

Given a training set $\mathcal{D}$, Bayesian learning estimates the predictive uncertainty $P(y^*|x^*, \mathcal{D})$, which is the marginalization of the posterior distribution $P(\boldsymbol{z}|\mathcal{D})$:

$$P(y^*|x^*, \mathcal{D}) = \int P(y^*|x^*, \boldsymbol{z}) P(\boldsymbol{z}|\mathcal{D}) \, d\boldsymbol{z}, \quad (7)$$

where $\boldsymbol{z}$ denotes unknown Bayesian variables, e.g., $\boldsymbol{z} := \{\mathbf{W}, b\}$ in SIKA-GP.

Using integration to compute the exact posterior $P(\boldsymbol{z}|\mathcal{D})$ is intractable in practice, and the posterior also may not be Gaussian. The BNN translation of the sparse GP in (6) provides us with an approximation of the underlying GP prior. To retain the same desired sparse properties in inference, we adopt variational inference (VI) to approximate the exact posterior $P(\mathbf{W}, b|\mathcal{D})$ by a variational distribution $q(\mathbf{W}, b)$. In particular, we select a variational family of mean-field Gaussian distributions on random weights $\mathbf{W}$ and bias $b$:

$$q(\mathbf{W}) = \mathcal{N}(\mathbf{m_W}, \text{diag}(\mathbf{s_W})), \; q(b) = \mathcal{N}(m_b, \sigma_b^2), \quad (8)$$

where $\mathbf{m_W}$ denotes the variational mean with width $DM$, and $\text{diag}(\mathbf{s_W})$ represents the variational covariance restricted to the diagonal entries given by $\mathbf{s_W}$. By mean-field

assumption, $q(\boldsymbol{z}) = q(\mathbf{W})q(b)$. The variational posterior is then optimized by maximizing the evidence lower bound (ELBO):

$$\text{ELBO} = \mathbb{E}_{q(\boldsymbol{z})} \left[\log P(y|\mathbf{x}, \boldsymbol{z})\right] - \text{KL}\left[q(\boldsymbol{z})\|p(\boldsymbol{z})\right]. \quad (9)$$

## 5. Tensor-based Sparse Indexing for Accelerated Inference of SIKA-GP

In this section, we present the details of the accelerated inference of SIKA-GP with the Laplace kernel. The explicit representation in Theorem 4.1 fully determines the sparsity pattern of $\phi(x)$ for each given $x$. In addition to the globally activated basis $\psi_{01}$ and $\psi_{02}$, there is at most one activated basis $\psi_{lm}$ for each $l = 1, \ldots, L$. Therefore, only a small subset of $\mathbf{W}$ is necessary to compute $\Phi(\mathbf{x})\mathbf{W}$. To facilitate efficient inference, we design the computation pipelines using a tensor representation.

### 5.1. Sparse indexing of activated tensors

To make effective use of this sparse structure efficiently on GPU, we define the inducing points and $\psi_{lm}$ in a dyadic order such that the index of activated $\mathbf{W}$, when multiplied by the sparsely activated $\Phi(\mathbf{x})$, can be determined in a parallelized manner. We assume that the data input is stored as a tensor $\mathbf{X} \in \mathbb{R}^{B \times D}$ in a mini-batch of size $B$.

**Dyadic inducing points**: Define the sets of dyadic points $\mathbf{U}_l$ with increasing order $l = 1, \ldots, L$ as a tensor

$$\mathbf{U}_l := \left[m \cdot 2^{-l} : m \in \{1, 3, 5, \ldots, 2^l - 1\}\right]. \quad (10)$$

Each $\mathbf{U}_l$ consists of $2^{l-1}$ inducing points, and the entire set of inducing points is

$$\mathbf{U}^{[L]} := [\mathbf{U}_0, \mathbf{U}_1, \ldots, \mathbf{U}_L], \quad (11)$$

**Dyadic Sorting algorithm:** 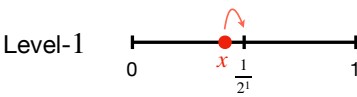

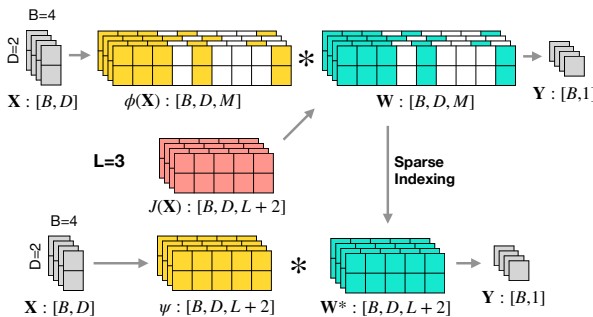

*Figure 4.* Parallelized lightweight forward process of SIKA-GP using TSI algorithm. "$*$" denotes the inner product (i.e., convolution linear layer) on the last two dimensions $[D, M]$.

*Figure 3.* An example of Sparse Indexing ($L = 3$).

where $\mathbf{U}_0 := [0, 1]$ denotes the boundary points. The *dyadic* $\phi(x)$ is therefore given by

$$\phi(x) = [\psi_{01}, \psi_{02}, \boldsymbol{\psi}_1, \boldsymbol{\psi}_2 \ldots, \boldsymbol{\psi}_L], \tag{12}$$

$$\boldsymbol{\psi}_l := [\psi_{l1}, \psi_{l3}, \ldots, \psi_{l(2^l - 1)}], \; l = 1, \ldots, L. \tag{13}$$

For example, the set of level-2 dyadic ordered inducing points is given by $\mathbf{U}^{[2]} = \left[0, 1, \frac{1}{2}, \frac{1}{4}, \frac{3}{4}\right]$.

**Corollary 5.1.** *The only activated function $\psi_{lm}$ in $\boldsymbol{\psi}_l$ is associated with the closest point $u_{lm} := m \cdot 2^{-l} \in \mathbf{U}_l$ to the input $x$ for $l = 1, \ldots, L$.*

Theorem 4.1 and Corollary 5.1 indicate that the activated $\phi(x) \in \mathbb{R}^M$ can be computed in a closed form after indexing the closest $u_{lm}$ in each order $l = 1, 2, \ldots, L$. An example is given in Figure 3. This indexing procedure can be performed in parallel using the following Tensorized Sparse Indexing (TSI) algorithm.

**Tensorized Sparse Indexing**: Given the input stored as a mini-batch tensor $\mathbf{X} \in \mathbb{R}^{B \times D}$, where $B$ is the batch size and $D$ is the feature dimension, define a length-$L$ tensor vector of the powers of 2 with increasing orders

$$\mathbf{r} := [2^1, 2^2, \ldots, 2^L] \in \mathbb{N}^{1 \times L}. \tag{14}$$

Given the input $\mathbf{X}$, $\phi(\mathbf{X})$ is a tensor matrix of dimension $(B \times D \times M)$. In SIKA-GP, the activated indices associated with $[\psi_1, \ldots, \psi_L]$ are

$$\mathbf{t} = \left\lfloor \frac{\lceil \mathbf{X} \odot \mathbf{r} \rceil + 1}{2} \right\rfloor \in \mathbb{N}^{B \times D \times L}, \tag{15}$$

where $\mathbf{X} \odot \mathbf{r}$ denotes the element-wise multiplication that broadcasts $x \odot \mathbf{r}$ throughout the whole batch dimension $B$ and the feature dimension $D$.

The activated tensor indices in $\phi(\mathbf{X})$ are, therefore, obtained by offset $\mathbf{t}$ and concatenated with the first two indices (two globally activated $\psi_{01}, \psi_{02}$)

$$J(\mathbf{X}) = \left[ \mathbf{1}, \; \mathbf{2}, \; \mathbf{t} + \underbrace{(\mathbf{r}/2 + 1)}_{\text{offset from } \psi \text{ to } \phi} \right]. \tag{16}$$

Since only indices of $\mathbf{W}$ that correspond to activated $\psi_{lm}$ are necessary for $\Phi(\mathbf{X})\mathbf{W}$, $J(\mathbf{X})$ is used to squeeze the entire weight matrix $\mathbf{W}$ as

$$\mathbf{W}^* := \mathbf{W}[J(\mathbf{X})] \in \mathbb{R}^{B \times D(L+2)} \tag{17}$$

for scalar output. For $C$ outputs, an additional channel dimension is used.

Note that all the operations in TSI support batch-wise parallelization, which enables efficient and GPU-friendly inference. This design enables a *lightweight* forward propagation and scalable training by reducing redundant kernel evaluations while maintaining expressive capacity. A parallelized forward process using TSI is illustrated in Figure 4.

Instead of sampling the entire vector $\mathbf{W} \in \mathbb{R}^{DM}$, we only need to sample $\mathbf{W}^* \in \mathbb{R}^{D(L+2)}$ as the activated rows of $\mathbf{W}$ are deterministically associated with the activated $\psi_{lm}$. By the mean-field VI, the variational distribution of $\mathbf{W}^*$ can be efficiently obtained as:

$$q(\mathbf{W}^*) = \mathcal{N}(\mathbf{m}_{\mathbf{W}}^*, \text{diag}(\mathbf{s}_{\mathbf{W}}^*)), \tag{18}$$

where $\mathbf{m}_{\mathbf{W}}^* = \mathbf{m}_{\mathbf{W}}[J(\mathbf{x})], \; \mathbf{s}_{\mathbf{W}}^* = \mathbf{s}_{\mathbf{W}}[J(\mathbf{x})]$.

During training, we optimize the ELBO on mini-batches of $\mathbf{X}$ with a total of $S$ random $\mathbf{W}^*$ samples:

$$\mathcal{L}(\theta) = -\frac{1}{S} \sum_{s=1}^{S} \log P(\mathbf{y} | \mathbf{X}, \mathbf{W}_s^*, b_s)$$

$$+ \text{KL}\left[q(\mathbf{W}, b) \| p(\mathbf{W}, b)\right], \tag{19}$$

$$\mathbf{W}_s^* = \mathbf{m}_{\mathbf{W}}^* + \mathbf{s}_{\mathbf{W}}^* \odot \boldsymbol{\epsilon}_s, \; \boldsymbol{\epsilon}_s \sim \mathcal{N}(\mathbf{0}, \boldsymbol{I}_{L+2}), \tag{20}$$

$$b_s = m_b + \sigma_b \cdot \epsilon_s, \; \epsilon_s \sim \mathcal{N}(0, 1). \tag{21}$$

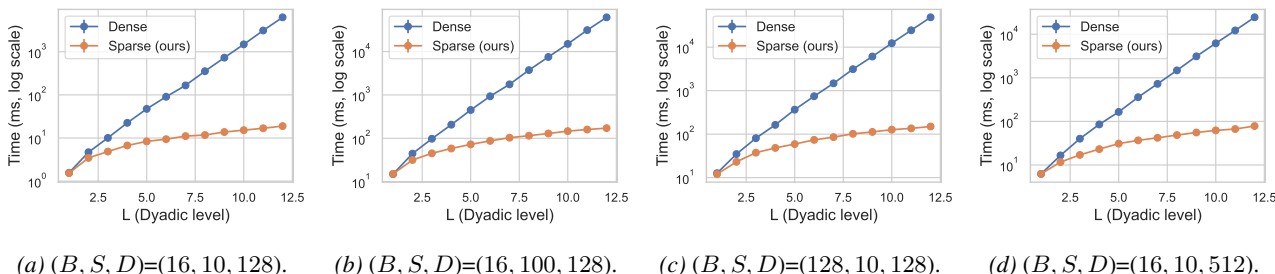

*(a)* $(B, S, D)$=(16, 10, 128).    *(b)* $(B, S, D)$=(16, 100, 128).    *(c)* $(B, S, D)$=(128, 10, 128).    *(d)* $(B, S, D)$=(16, 10, 512).

*Figure 5.* **CPU** inference time of SIKA-GP. $B$: batch size; $S$: the number of MC samples; $D$: the dimension of features.

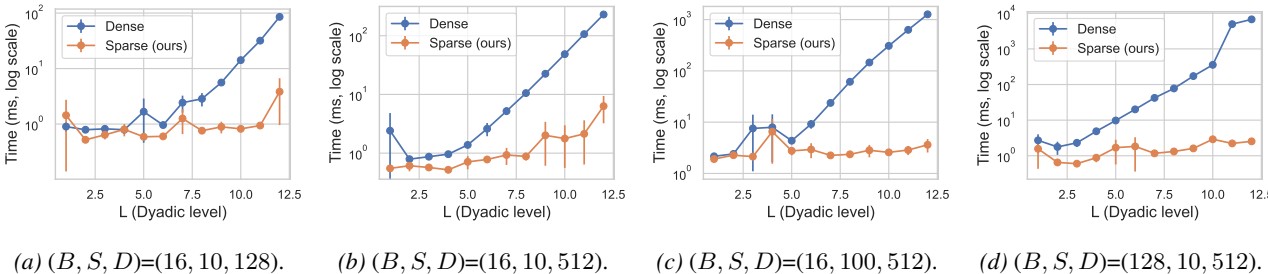

*(a)* $(B, S, D)$=(16, 10, 128).    *(b)* $(B, S, D)$=(16, 10, 512).    *(c)* $(B, S, D)$=(16, 100, 512).    *(d)* $(B, S, D)$=(128, 10, 512).

*Figure 6.* **CUDA** inference time of SIKA-GP. $B$: batch size; $S$: the number of MC samples; $D$: the dimension of features.

During inference, we predict for a new $x^*$ by similarly drawing $S^*$ samples from the learned variational distribution:

$$\mathbb{E}_{q(\mathbf{W}^*, b)} \left[ P(y^*|x^*, \mathbf{W}^*, b) \right] \approx \frac{1}{S^*} \sum_{s=1}^{S^*} P\left(y^*|x^*, \mathbf{W}_s^*, b_s\right).$$

Figure 4 illustrates the accelerated inference, which reduces the cost of sampling and multiplication from $O(M)$ to $O(\log M)$ with $M = 2^L + 1$ inducing points.

**Scalability**. We summarize the computational complexity of our proposed SIKA-GP compared to other approximated GPs in Table 1. To make a fair comparison, the number of inducing points in SVGP and KISS-GP are also chosen to be $M = 2^L + 1$. We denote the number of data points to be computed as $N$ and the number of samples to be drawn as $S$ in both training and inference. However, for more complex or multitasking GPs, it may be necessary to use a larger number of inducing points in both SVGP and KISS-GP, since this quantity is tied to the dimensionality of the GP inputs. Conversely, as the dataset becomes more intricate and higher-dimensional, it is essential to increase the number of inducing points and Monte Carlo samples.

# 6. Experiments

We evaluate the proposed SIKA-GP on multiple synthetic benchmarks and real datasets. We first analyze the time complexity of SIKA-GP and then evaluate SIKA-GP in multiple BDL applications, including real datasets for regression, vision, and language models.[3]

---

*Table 1.* Computational complexity of different one-layer GPs. $N$: the number data points; $M$: the number of inducing points; $S$: the number of MC samples used for training/inference; $D$: the dimension of data features.

| | **Training** | **Inference** |
|---|---|---|
| SVGP | $O(SDNM + NM^2 + M^3)$ | $O(SNM^2)$ |
| KISS-GP | $O(SDNM + DM^{\frac{3}{D}})$ | $O(SDM^{1+\frac{1}{D}})$ |
| DAK | $O(SDNM)$ | $O(SDM)$ |
| SIKA-GP (ours) | $O(SDN \log M + DM)$ | $O(SD \log M)$ |

## 6.1. Time analysis

Since SIKA-GP can be recast as a BNN layer with sparsely activated kernel activation $\phi(x) = K(x, \mathbf{U})\mathbf{P}_{\mathbf{U}}^{-\top}$ and Bayesian random weight $\mathbf{w}$, we benchmark the inference time of SIKA-GP compared to a one-layer BNN with the same width of weight.

We compare the inference time of Dense BNN versus Sparse (ours) indexing of SIKA-GP at varying dyadic levels $L$, with different batch sizes $B$, number of MC samples $S$ and feature dimension $D$. In both CPU and CUDA benchmarks, the proposed Sparse method consistently demonstrates appreciable reductions in inference time, especially as the dyadic level $L$ increases.

In Figure 5, the CPU time for Dense inference grows exponentially with $L$, achieving a cost of several orders of magnitude higher than that of $L > 5$. In contrast, our Sparse inference increases much more slowly, remaining nearly linear in $L$. Across different settings of $(B, S, D)$, our sparse inference consistently outperforms the Dense baseline, with relative gains more pronounced for larger $D$

and higher $L$.

In Figure 6, we show the impact of GPU parallelization on Dense and Sparse inference. Although both methods benefit from CUDA, Dense inference still suffers from exponential scaling with $L$. However, Sparse inference remains nearly linear and less than 10 ms, demonstrating that the approximation significantly improves GPU utilization. Even in large batch and dimensional setups (e.g. $B = 128, D = 512$), the Sparse approach maintains a millisecond-level inference time, whereas Dense inference rapidly grows to hundreds or thousands of milliseconds.

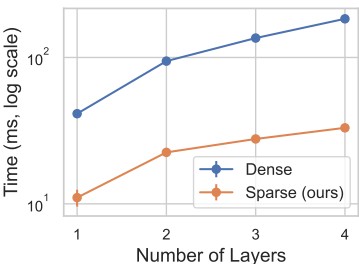

*Figure 7.* CUDA inference time of deep SIKA-GP.

Figure 7 shows the inference time of the deep SIKA-GP with multiple layers. We set $(B, D, S) = (64, 128, 10)$ with $M = 129$ inducing points. The dense baseline grows rapidly with depth, exceeding 100 ms by more than 2 layers, while our sparse approximation scales much more gracefully and remains below 30 ms. The growing gap shows the appropriateness of our proposed sparse inference for deep and hierarchical Bayesian architectures.

## 6.2. DGP in UCI Regression

We evaluate SIKA-GP on UCI regression datasets using a DGP with 2 hidden layers of SIKA-GP. The dyadic level of each layer is set to $L = 7$. The training employs the Adam optimizer for 100 epochs, using 10 MC samples, and a batch size of 512. We refer to DGP with our proposed SIKA-GP method as DGP-SIKA, while we refer to the dense DGP with no acceleration as DGP. A detailed training recipe is provided in Section C.2.

*Table 2.* Predictive performance and runtime on UCI regression benchmarks. DGP-SIKA achieves comparable predictive accuracy while reducing time cost across datasets.

| Datasets | Models | RMSE ↓ | NLPD ↓ | Time(s) ↓ |
|---|---|---|---|---|
| Gas | DGP | 0.54±0.01 | 1.09±0.12 | 19.43±1.66 |
| | DGP-SIKA | 0.53±0.03 | 1.07±0.11 | 2.79±0.04 |
| Kin40K | DGP | 0.09±0.01 | 0.07±0.01 | 33.25±0.56 |
| | DGP-SIKA | 0.09±0.01 | 0.07±0.01 | 15.76±0.51 |
| Protein | DGP | 0.67±0.04 | 0.94±0.04 | 56.50±1.14 |
| | DGP-SIKA | 0.68±0.04 | 0.94±0.04 | 28.92±0.63 |

On the UCI benchmarks (Gas, Kin40K, and Protein), DGP-

SIKA consistently matches the predictive performance of dense DGPs in both RMSE and NLPD. More importantly, DGP-SIKA achieves significant efficiency gains: Runtime per epoch is reduced by up to $7\times$ on Gas and about $2\times$ on Kin40K and Protein. These results show that SIKA-GP preserves predictive accuracy while significantly improving scalability, making DGPs with SIKA more practical for real-world regression tasks.

## 6.3. DKL in Image Classification

To evaluate the application of SIKA-GP in highly structured and high-dimensional image data, such as MNIST (LeCun et al., 1998), CIFAR-10/100 (Krizhevsky et al., 2009), we incorporate SIKA-GP into DKL models, where the inputs to the GP are high-dimensional features learned by a DNN.

**Setup**. Every model incorporates the same DNN head for feature extraction, followed by either a linear output layer for the NN model or the respective last-layer GPs for SVDKL and DAK. MNIST employs a simple CNN with 64 extracted features; CIFAR-10 employs `ResNet-18` with 128 extracted features; CIFAR-100 employs `ResNet-34` with 512 output features. In every model, the GP layers incorporate 129 inducing points with a dyadic level of $L = 7$. In classification, the outputs of the last layer are followed by a Softmax layer to normalize the output to a probability distribution, and we perform MC sampling with $S = 20$ to estimate the expected likelihood in ELBO. We refer to our Sparse Indexing method as DKL-SIKA, while we refer to the original dense DAK as DAK. We report the average results over 3 runs for all experiments, and the best results are highlighted in bold. The detailed training recipe is provided in Section C.3.

**Results**. Across MNIST, CIFAR-10, and CIFAR-100, DKL-SIKA consistently achieves competitive or improved predictive accuracy and uncertainty calibration while improving computational efficiency. In MNIST, DKL-SIKA reduces the training time to 10.09 s/epoch, representing a $1.3\times$ speedup over DAK (13.73s) and $1.6\times$ faster than SVDKL (16.46s). In CIFAR-10, DKL-SIKA achieves the highest accuracy (94.78%) and the lowest calibration error (ECE = 4.06%), offering a slight improvement over both DAK and SVDKL. In terms of efficiency, DKL-SIKA reduces the training time to 17.26 s/epoch, about $2\times$ faster than DAK and $3\times$ than SVDKL. Test-time inference is also faster, with DKL-SIKA (4.50 s) outperforming both DAK (6.22 s) and SVDKL (6.08 s). In the more challenging CIFAR-100 benchmark, DKL-SIKA maintains slightly better accuracy and ECE than DAK, while reducing training and inference costs by more than $3\times$. In settings where standard SVDKL exhibits poor convergence, fine-tuned DKL-SIKA with a pretrained `ResNet-34` backbone achieves the highest accuracy, lowest NLL, and best ECE among all baselines,

*Table 3.* Evaluation of accuracy, uncertainty quality, and training/inference efficiency on MNIST, CIFAR-10, and CIFAR-100. Accuracy (**ACC**) and Expected Calibration Error (**ECE**) are reported in percentages. Best results are highlighted in **boldface**.

| Dataset | Model | ACC(%) ↑ | NLL ↓ | ECE(%) ↓ | ELBO(%) ↑ | Train Time(s/epoch) ↓ | Infer Time(s) ↓ |
|---|---|---|---|---|---|---|---|
| MNIST | SVDKL | $98.17 \pm 0.18$ | $0.09 \pm 0.03$ | $1.87 \pm 0.26$ | $-0.16 \pm 0.03$ | $16.46 \pm 0.30$ | $1.81 \pm 0.14$ |
| | DAK | $99.06 \pm 0.02$ | $0.03 \pm 0.00$ | $\mathbf{0.50 \pm 0.03}$ | $-0.03 \pm 0.00$ | $13.73 \pm 0.40$ | $1.42 \pm 0.06$ |
| | DKL-SIKA (ours) | $\mathbf{99.06 \pm 0.01}$ | $\mathbf{0.03 \pm 0.00}$ | $0.53 \pm 0.03$ | $-0.03 \pm 0.01$ | $\mathbf{10.09 \pm 0.15}$ | $\mathbf{1.00 \pm 0.09}$ |
| CIFAR-10 | SVDKL | $94.28 \pm 0.26$ | $\mathbf{0.26 \pm 0.01}$ | $4.10 \pm 0.08$ | $-0.83 \pm 0.01$ | $46.90 \pm 1.61$ | $6.08 \pm 0.51$ |
| | DAK | $94.53 \pm 0.32$ | $0.28 \pm 0.03$ | $4.34 \pm 0.42$ | $-0.20 \pm 0.04$ | $35.04 \pm 0.49$ | $6.22 \pm 0.21$ |
| | DKL-SIKA (ours) | $\mathbf{94.78 \pm 0.31}$ | $0.27 \pm 0.02$ | $\mathbf{4.06 \pm 0.23}$ | $-0.20 \pm 0.02$ | $\mathbf{17.26 \pm 0.28}$ | $\mathbf{4.50 \pm 0.13}$ |
| CIFAR-100 | ResNet-34 | $75.28 \pm 0.50$ | $1.03 \pm 0.02$ | $8.72 \pm 0.36$ | — | $8.76 \pm 0.03$ | $1.98 \pm 0.11$ |
| | DAK | $76.61 \pm 0.16$ | $\mathbf{1.23 \pm 0.02}$ | $5.55 \pm 0.45$ | $-2.41 \pm 0.03$ | $102.57 \pm 1.95$ | $5.74 \pm 4.78$ |
| | DKL-SIKA (ours) | $\mathbf{76.94 \pm 0.41}$ | $1.25 \pm 0.04$ | $\mathbf{4.31 \pm 0.33}$ | $\mathbf{-1.23 \pm 0.20}$ | $\mathbf{36.31 \pm 0.58}$ | $\mathbf{4.22 \pm 1.18}$ |
| | SVDKL-FT | $76.28 \pm 0.18$ | $0.96 \pm 0.01$ | $3.69 \pm 0.20$ | $-7.43 \pm 0.02$ | $41.69 \pm 0.80$ | $13.44 \pm 0.50$ |
| | DAK-FT | $76.13 \pm 0.10$ | $0.99 \pm 0.05$ | $4.18 \pm 0.33$ | $-9.66 \pm 0.05$ | $35.62 \pm 0.59$ | $9.66 \pm 0.78$ |
| | DKL-SIKA-FT | $\mathbf{76.61 \pm 0.40}$ | $\mathbf{0.94 \pm 0.01}$ | $\mathbf{3.45 \pm 0.45}$ | $\mathbf{-2.54 \pm 0.37}$ | $\mathbf{20.08 \pm 0.45}$ | $\mathbf{7.07 \pm 1.56}$ |

*Table 4.* OOD performance and calibration on CLINC150. Best results are highlighted in **boldface**.

| Model | $\text{ACC}_{\text{OOD}}$(%) ↑ | $\text{NLL}_{\text{OOD}}$ ↓ | $\text{ECE}_{\text{OOD}}$(%) ↓ | AUROC ↑ | AUPRC ↓ | Time(min/epoch) ↓ |
|---|---|---|---|---|---|---|
| SVDKL | $88.24 \pm 0.24$ | $0.62 \pm 0.01$ | $\mathbf{5.27 \pm 0.28}$ | $0.8413 \pm 0.0085$ | $0.0228 \pm 0.0082$ | $9.35 \pm 0.46$ |
| DAK | $87.70 \pm 0.18$ | $0.56 \pm 0.02$ | $5.43 \pm 0.30$ | $0.8486 \pm 0.0082$ | $0.0197 \pm 0.0014$ | $6.22 \pm 0.15$ |
| DKL-SIKA (ours) | $\mathbf{88.09 \pm 0.37}$ | $\mathbf{0.54 \pm 0.01}$ | $5.41 \pm 0.22$ | $\mathbf{0.8590 \pm 0.0083}$ | $\mathbf{0.0190 \pm 0.0016}$ | $\mathbf{3.41 \pm 0.05}$ |

*Table 5.* ID performance on CLINC150.

| Model | $\text{ACC}_{\text{ID}}$(%) ↑ | $\text{NLL}_{\text{ID}}$ ↓ | $\text{ECE}_{\text{ID}}$(%) ↓ |
|---|---|---|---|
| SVDKL | $95.03 \pm 0.37$ | $0.26 \pm 0.01$ | $2.34 \pm 0.30$ |
| DAK | $94.65 \pm 0.22$ | $0.26 \pm 0.01$ | $2.12 \pm 0.08$ |
| DKL-SIKA (ours) | $\mathbf{95.03 \pm 0.08}$ | $\mathbf{0.24 \pm 0.00}$ | $\mathbf{2.05 \pm 0.21}$ |

while maintaining the best efficiency.

In general, these results demonstrate that DKL-SIKA maintains strong predictive and uncertainty performance while offering substantial gains in training and inference efficiency. The lightweight forward pass on $\mathbf{W}^*$ allows for more efficient and adequate MC sampling to better estimate the ELBO, potentially improving uncertainty quantification.

### 6.4. Transformer-based Language Models

Although GPs have been extensively studied in small-to medium-scale settings, their application to modern transformer-based language models (TLMs) remains limited due to the prohibitive computational cost of kernel inference. We evaluate SIKA-GP on the CLINC150 (Larson et al., 2019) intent classification benchmark, which contains 150 in-distribution (ID) intents and a designated out-of-distribution (OOD) class. Models are trained only on ID intents, while OOD detection is evaluated post hoc using predictive uncertainty.

**Setup**. We use `DistilBERT` as the feature encoder and apply GP layers as the final predictive layer, enabling structured uncertainty estimation on top of transformer repre-

sentations. We report accuracy, ECE, and NLL in the ID validation set and evaluate AUROC and AUPRC using predictive entropy in the OOD test set. The detailed setting can be found in Section C.4.

**Results**. Tables 4 and 5 show that DKL-SIKA consistently matches or improves predictive accuracy compared to SVDKL and DAK, while achieving lower NLL and ECE, indicating better calibrated predictions without sacrificing classification performance. Under distribution shift, DKL-SIKA yields the lowest NLL and the highest AUROC, demonstrating more reliable uncertainty estimates for separating ID and OOD intents. All GP baselines use the same predictive-uncertainty scoring protocol; the advantage of SIKA-GP is that its cheaper stochastic forward pass makes it practical to average more posterior samples, which reduces the noise of entropy or mutual-information based OOD scores. In terms of computational complexity, these improvements come with a markedly lower training cost, reducing the time per epoch by more than $2\times$ relative to SVDKL. Overall, the results indicate that SIKA-GP provides an efficient and scalable layer for uncertainty estimation when combined with TLMs, filling a gap that earlier GP approaches did not address.

### 6.5. Ablation Study

We perform two ablations to identify: 1) whether the Laplace kernel restriction causes a large predictive trade-off; 2) how sensitive SIKA-GP is to the dyadic level $L$ or equivalently the number of inducing points $M = 2^L + 1$.

*Table 6.* Results on CIFAR-100 with different choices of kernels. Best values are shown in bold.

| Kernel | ACC ↑ | ECE / NLL ↓ | Train / Infer ↓ |
|---|---|---|---|
| RBF (DAK) | **77.43** | 5.36 / 0.98 | 37.38 / 9.78 |
| Laplace (DAK) | 76.13 | 4.18 / 0.99 | 35.62 / 9.66 |
| Laplace (SIKA) | 76.61 | **3.45 / 0.94** | **20.08 / 7.07** |

**Choices of kernels.** The kernel ablation compares an RBF kernel, a Laplace kernel with dense inference, and the proposed SIKA on CIFAR-100. All three kernel choices are used as the last GP layer under the same DKL setting. This isolates whether the computationally favorable Laplace structure introduces a prohibitive predictive penalty in a deep-feature setting. As shown in Table 6, the dense RBF achieves the highest accuracy, but it has higher training and inference cost. Replacing RBF by a dense Laplace kernel slightly reduces accuracy, while applying SIKA to the Laplace kernel recovers part of the accuracy, improves calibration, and substantially reduces training and inference time. The results support our intended claim: the Laplace assumption is a modeling trade-off, but in the deep kernel setting, the trade-off can be small relative to the computational gain.

*Table 7.* CIFAR-10 performance under different dyadic levels $L$ ($M = 2^L + 1$). Best values are shown in bold.

| $L$ | 1 | 2 | 4 | 7 | 10 |
|---|---|---|---|---|---|
| $M = 2^L + 1$ | 3 | 5 | 17 | 129 | 1025 |
| Active bases | 3 | 4 | 6 | 9 | 12 |
| ACC | 92.59 | 94.56 | 94.59 | 94.76 | **94.78** |
| ECE | 5.79 | 4.17 | 4.78 | 4.10 | **4.06** |
| NLL | 0.420 | 0.279 | 0.391 | 0.281 | **0.270** |

**Choices of the dyadic level.** We next vary the dyadic level $L$ on CIFAR-10. Since $M = 2^L + 1$, this ablation covers settings from 3 to 1025 inducing points; however, each input activates only $L + 2$ basis functions. The results indicate mild sensitivity to the number of inducing points once $L \geq 2$: accuracy improves sharply from $L = 1$ to $L = 2$ and then remains stable, while ECE and NLL also stay competitive at larger levels. Importantly, increasing $M$ from 129 to 1025 only increases the number of active basis functions from 9 to 12, which explains why larger inducing grids can be used avoiding dense $O(M)$ inference.

## 7. Conclusion

We presented a GP with sparse inducing kernel approximations (SIKA-GP) that accelerates GP inference by exploiting the sparsely activated basis functions. SIKA-GP reduces the complexity of the inference to $O(\log M)$, where $M$ is the number of inducing points designed on a dyadic grid.

Our approach maintains predictive accuracy, offers significant efficiency improvements, and seamlessly extends to deep kernel models due to its tensorized sparse indexing on equivalent BNN representations. Experimental results highlight SIKA-GP as a practical and scalable approach to bringing GPs into modern Bayesian deep learning.

**Limitations**: 1) The underlying structured sparsity of the basis functions partly arises from the restrictive Laplace kernel, although this limited expressiveness can be alleviated through deep neural models. 2) Despite its efficiency and stability, SIKA-GP trades some flexibility in learning arbitrary inducing locations for a structured fixed grid.

**Future work** will explore adaptive or hybrid dyadic grids, extensions of the sparse execution pipeline to broader kernel classes, and integration with parameter-efficient fine-tuning methods such as LoRA for modern Bayesian large-scale models.

## Acknowledgements

The work of W. Zhao and C. Tian was supported in part by the National Science Foundation via grants DMS-2312173 and ECCS-2433631. The authors gratefully acknowledge the computing infrastructure provided by the Department of Electrical & Computer Engineering at Texas A&M University, and the reviewers' constructive feedback.

## Impact Statement

This work aims to make GP-based uncertainty estimation more computationally practical in modern deep learning pipelines. There are many potential societal consequences of our work, especially in advancing the field of Gaussian Processes and trustworthy AI/ML.

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

# A. Proof of Theorems

## A.1. Proof of Theorem 4.1

Our proof needs the notion of the reproducing kernel Hilbert space (RKHS) generated by $K$, denoted as $\mathcal{N}_K[0, 1]$. Denote the inner product and the norm of this space by $\langle \cdot, \cdot \rangle_{\mathcal{N}}$ and $\| \cdot \|_{\mathcal{N}}$, respectively. Recall that $\mathcal{N}_K[0, 1]$ is defined as the completion of the function space

$$\left\{ \sum_{j=1}^{N} \beta_j K(\cdot, x_j) : N \in \mathbb{N}, \beta_j \in \mathbb{R}, x_j \in [0, 1] \right\} \tag{22}$$

under the inner product

$$\left\langle \sum_{j=1}^{N} \beta_j K(\cdot, x_j), \sum_{k=1}^{N'} \beta'_k K(\cdot, x'_k) \right\rangle_{\mathcal{N}} = \sum_{j=1}^{N} \sum_{k=1}^{N'} \beta_j \beta'_k K(x_j, x'_k). \tag{23}$$

A remarkable property of the reproducing kernel Hilbert space is the reproducing property, namely,

$$f(x) = \langle f, K(\cdot, x) \rangle_{\mathcal{N}}, \tag{24}$$

for all $f \in \mathcal{N}_K[0, 1]$ and $x \in [0, 1]$.

**Theorem 1**(Restated). Given the Laplace kernel $K(x, y) = \exp(-\theta|x - y|)$ and inducing points $\mathbf{U} := \{0 \cdot 2^{-L}, 1 \cdot 2^{-L}, \ldots, 2^L \cdot 2^{-L}\}$, there is a sparsely activated BNN in terms of $f(x) = K(x, \mathbf{U})[K(\mathbf{U}, \mathbf{U})]^{-1} f(\mathbf{U}) := \phi(x)\mathbf{w}$ with a set of basis functions $\phi(x) = \{\psi_{lm} : [0, 1] \to \mathbb{R}\}$ and $\mathbf{w} \sim \mathcal{N}(0, \mathbf{I})$, given by

$$\psi_{01}(x) := \frac{\exp\{-\theta x\} + \exp\{-\theta(1 - x)\}}{\sqrt{2(1 + \exp\{-\theta\})}},$$

$$\psi_{02}(x) := \frac{\exp\{-\theta x\} - \exp\{-\theta(1 - x)\}}{\sqrt{2(1 - \exp\{-\theta\})}},$$

and

$$\psi_{lm}(x) := \begin{cases} \sqrt{\dfrac{2}{\sinh(2^{1-l}\theta)}} \, \sinh(\theta(x - (m - 1)2^{-l})), \\ \qquad \text{if } (m - 1)2^{-l} \le x \le m2^{-l}, \\[2mm] \sqrt{\dfrac{2}{\sinh(2^{1-l}\theta)}} \, \sinh(\theta((m + 1)2^{-l} - x)), \\ \qquad \text{if } m2^{-l} \le x \le (m + 1)2^{-l}, \\[2mm] 0, \text{ if } x \notin [(m - 1)2^{-l}, (m + 1)2^{-l}], \end{cases}$$

for $l = 1, 2, \ldots, L$, and $m = 1, 3, \ldots, 2^l - 1$.

*Proof of Theorem 1.* It suffices to prove that there exists a matrix $\mathbf{P_U}$ satisfying $K(\mathbf{U}, \mathbf{U}) = \mathbf{P_U} \mathbf{P_U}^\top$, such that $\phi(x) := K(x, \mathbf{U})\mathbf{P_U}^{-\top}$ consists of all $\psi_{lm}$'s.

To show this, consider $\phi_\mathbf{P}(\cdot) := K(\cdot, \mathbf{U})\mathbf{P}^{-\top}$ with an invertible $\mathbf{P}$. The Gram matrix of $\phi_\mathbf{P}(\cdot)$ under $\langle \cdot, \cdot \rangle_{\mathcal{N}}$ is

$$\langle \phi_\mathbf{P}^\top, \phi_\mathbf{P} \rangle_{\mathcal{N}} = \mathbf{P}^{-1} \langle K(\cdot, \mathbf{U})^\top, K(\cdot, \mathbf{U}) \rangle_{\mathcal{N}} \mathbf{P}^{-\top} = \mathbf{P}^{-1} K(\mathbf{U}, \mathbf{U}) \mathbf{P}^{-\top}.$$

This implies that $\langle \phi_\mathbf{P}^\top, \phi_\mathbf{P} \rangle_{\mathcal{N}} = \mathbf{I}$ if and only if $K(\mathbf{U}, \mathbf{U}) = \mathbf{P}\mathbf{P}^\top$. Consequently, $K(\mathbf{U}, \mathbf{U}) = \mathbf{P}\mathbf{P}^\top$ if and only if $\phi_\mathbf{P}$ is an orthonormal basis of $\mathbf{V} := \text{span}\{K(\cdot, 0 \cdot 2^{-L}), K(\cdot, 1 \cdot 2^{-L}), \ldots, K(\cdot, 2^L \cdot 2^{-L})\}$ under $\langle \cdot, \cdot \rangle_{\mathcal{N}}$.

Therefore, it suffices to prove that $\psi_{lm}$ forms an orthonormal basis of $\mathbf{V}$ under $\langle \cdot, \cdot \rangle_{\mathcal{N}}$. Such a result can be verified via direct calculations or a more principled method of construction introduced in Section B that works for general Gauss-Markov covariances. $\square$

## A.2. Proof of Theorem 5.1

**Corollary 1**(Restated). *The only activated function $\psi_{lm}$ in $\psi_l := \{\psi_{lm} : m \in \{1, 3, \ldots, 2^l - 1\}\}$ is associated with the closest point $u_{lm} := m \cdot 2^{-l} \in \mathbf{U}_l$ to the input $x$ for all $l = 1, \ldots, L$.*

*Proof of Corollary 1.* Let the inducing points be $\mathbf{U} := \{0 \cdot 2^{-L}, 1 \cdot 2^{-L}, \ldots, 2^L \cdot 2^{-L}\}$. Define the dyadic point sets $\mathbf{U}_l$ with order $l = 1, \ldots, L$ as

$$\mathbf{U}_l := \left[ m \cdot 2^{-l} : m \in \{1, 3, 5, \ldots, 2^l - 1\} \right]. \tag{25}$$

It is obvious that each $\mathbf{U}_l$ contains $2^{l-1}$ inducing points and we have

$$\mathbf{U}_i \bigcap \mathbf{U}_j = \emptyset, \text{ if } i \neq j. \tag{26}$$

Each $\psi_{lm}(x)$, $m = 1, 3, \ldots, 2^l - 1$, is supported in a local region $[(m-1)2^{-l}, (m+1)2^{-l}]$. Therefore, given a specific order $l \in \{1, 2, \ldots, L\}$, the activated $\psi_{lm*}(x)$ implies that

$$m^* = \underset{m \in \{1, 3, \ldots, 2^l - 1\}}{\arg\min} |x - m \cdot 2^{-l}|. \tag{27}$$

If there exists another $(m+2)2^{-l}$ or $(m-2)2^{-l}$ that is closer to $x$, then $x$ will be located in $(-\infty, (m-1)2^{-l}] \cup [(m+1)2^{-l}, +\infty)$ which implies that $\psi_{lm}(x) = 0$. Hence, if $\psi_{lm}(x) > 0$ for specific $l \in \{1, \ldots, L\}$ and $m \in \{1, 3, \ldots, 2^l - 1\}$, the closest inducing point in $\mathbf{U}_l$ is given by $u_{lm} = m \cdot 2^{-l}$.

Therefore, the *only* activated function $\psi_{lm}$ in $\psi_l$ is associated with the closest point $u_{lm} := m \cdot 2^{-l} \in \mathbf{U}_l$ to the input $x$ for $l = 1, \ldots, L$. $\qquad\square$

# B. Compact Supported Orthonormal Basis for Gauss-Markov Covariances

Let $K(x, y)$ be the covariance function of a continuous Gauss-Markov process on $[0, 1]$. Use the notation $a \wedge b := \min\{a, b\}$ and $a \vee b := \max\{a, b\}$. It is known that (Marcus & Rosen, 2006) $K(x, y)$ must admit the representation $K(x, y) = p(x \wedge y)q(x \vee y)$, for some functions $p(t) > 0$ and $q(t) > 0$ for all $t \in (0, 1)$. For instance, the Laplace correlation can be written as

$$K(x, y) = \exp(-\theta|x - y|) = \exp(\theta(x \wedge y))\exp(-\theta(x \vee y)).$$

## B.1. Template basis function

Our basis functions will be constructed using a "template basis function", denoted by $\phi_{a,b,c}(\cdot)$, for $0 \leq a < b < c \leq 1$. The function $\phi_{a,b,c}(\cdot)$ is defined by the following Theorem B.1.

**Theorem B.1.** *Given any $0 \leq a < b < c \leq 1$, there exists a unique function $\phi_{a,b,c}(\cdot)$ so that the following are true:*

1. *$\phi_{a,b,c}(x) = AK(x, a) + BK(x, b) + CK(x, c)$ for some $A, C \in \mathbb{R}$ and $B > 0$.*

2. *$\phi_{a,b,c}(x) = 0$ whenever $x \in [0, a] \cup [c, 1]$.*

3. *$\|\phi_{a,b,c}\|_{\mathcal{N}} = 1$.*

*Proof.* We will prove this theorem by constructing $A, B$ and $C$ so that the statements in the theorem hold, and show that such a construction is unique.

We note that, to ensure $\phi_{a,b,c}(a) = \phi_{a,b,c}(c) = 0$, we need

$$AK(a, a) + BK(a, b) + CK(a, c) = 0, \tag{28}$$
$$AK(a, c) + BK(b, c) + CK(c, c) = 0. \tag{29}$$

We will show that, once (28) and (29) are true, Statement 2 is fulfilled. First, we consider $x \in [0, a]$. We can assume that $0 < a$, because otherwise there is nothing to prove. Then, we know that $p(a) > 0$, and

$$
\begin{aligned}
&AK(x, a) + BK(x, b) + CK(x, c) \\
= &Ap(x \wedge a)q(x \vee a) + Bp(x \wedge b)q(x \vee b) + Cp(x \wedge c)q(x \vee c) \\
= &Ap(x)q(a) + Bp(x)q(b) + Cp(x)q(c) \\
= &\frac{p(x)}{p(a)}\{Ap(a)q(a) + Bp(a)q(b) + Cp(a)q(c)\} \\
= &\frac{p(x)}{p(a)}\{AK(a, a) + BK(a, b) + CK(a, c)\} = 0.
\end{aligned}
$$

The analogous result for $x \in [c, 1]$ can be derived in a similar manner.

Now we incorporate the norm condition $\|\phi_{a,b,c}\|_{\mathcal{N}} = 1$. Let

$$\mathbf{K} = \begin{pmatrix} K(a,a) & K(a,b) & K(a,c) \\ K(a,b) & K(b,b) & K(b,c) \\ K(a,c) & K(b,c) & K(c,c) \end{pmatrix}.$$

Then the norm condition can be represented by

$$(A, B, C)\mathbf{K}(A, B, C)^T = 1.$$

In view of (28) and (29), the above equation is reduced to

$$ABK(a,b) + B^2 K(b,b) + BCK(b,c) = 1. \tag{30}$$

We denote $(e_1, e_2, e_3) := I_3$, the $3 \times 3$ identity matrix. Then (28)-(30) is equivalent to

$$\mathbf{K}(AB, B^2, BC)^T = e_2.$$

Given the condition $B > 0$, the above equation has a unique solution

$$B = \sqrt{e_2^T \mathbf{K}^{-1} e_2}, A = e_1^T \mathbf{K}^{-1} e_2 / B, C = e_3^T \mathbf{K}^{-1} e_2 / B,$$

in which $B$ is indeed positive because $e_2^T \mathbf{K}^{-1} e_2 > 0$ as $K$ is positive definite. Then we complete the proof. $\qquad\square$

The next result is simple yet useful.

**Theorem B.2.** *Suppose $0 \le a < b < c \le 1$. Then for any $f \in \mathcal{N}_K[0,1]$ satisfying $f(a) = f(b) = f(c) = 0$, we have*

$$\langle \phi_{a,b,c}, f \rangle_{\mathcal{N}} = 0.$$

*Proof.* This result is a direct consequence of the reproducing property, as

$$\begin{aligned} \langle \phi_{a,b,c}, f \rangle_{\mathcal{N}} &= \langle AK(\cdot, a) + BK(\cdot, b) + CK(\cdot, c), f \rangle_{\mathcal{N}} \\ &= Af(a) + Bf(b) + Cf(c) = 0. \end{aligned}$$

$\qquad\square$

## B.2. Orthonormal basis

Note that each element of $x \in \mathbf{U} \setminus \{0, 1\}$ can be uniquely represented as $m2^{-l}$ for some odd number $m$ and $l \in \mathbb{N}_+$. We denote $T(x) = (l, m)$, and $L(x) = l$. Then for each $x \in \mathbf{U}$ and $T(x) = (l, m)$, define

$$\psi_{T(x)} = \psi_{l,m} := \phi_{x-2^{-l}, x, x+2^{-l}}.$$

We have the following theorem, which shows that $\{\psi_{T(x)} : x \in \mathbf{U} \setminus \{0, 1\}\}$ forms an orthonormal basis of a $2^L - 1$-dimensional subspace of $\mathbf{V} := \text{span}\{K(\cdot, 0 \cdot 2^{-L}), K(\cdot, 1 \cdot 2^{-L}), \dots, K(\cdot, 2^L \cdot 2^{-L})\}$.

**Theorem B.3.** *The following statements are true:*

1. $\langle \psi_{T(x)}, \psi_{T(x')} \rangle_{\mathcal{N}} = 1_{\{x = x'\}}$, *for $x, x' \in \mathbf{U} \setminus \{0, 1\}$.*

2. $\langle \psi_{T(x)}, K(\cdot, 0) \rangle_{\mathcal{N}} = \langle \psi_{T(x)}, K(\cdot, 1) \rangle_{\mathcal{N}} = 0$.

3. *Let $V_0 := \text{span}\{K(\cdot, 0), K(\cdot, 1)\}$. Suppose $\{\psi_{0,0}, \psi_{0,1}\}$ is an orthonormal basis of $V_0$. Then $\{\psi_{00}, \psi_{01}, \psi_{T(x)} : x \in \mathbf{U} \setminus \{0, 1\}\}$ forms an orthonormal basis of $\mathbf{V} := \text{span}\{K(\cdot, 0 \cdot 2^{-L}), K(\cdot, 1 \cdot 2^{-L}), \dots, K(\cdot, 2^L \cdot 2^{-L})\}$.*

*Proof.* First, we prove Statement 1. Let $(l, m) = T(x)$ and $(l', m') = T(x')$. When $m = m'$ and $l = l'$, this statement is ensured by the definition of the template basis functions. If $m \ne m'$ or $l \ne l'$, then either the supports of $\psi_{l,m}$ and $\psi_{l',m'}$ are non-overlapping, or the support of one function is contained within that of the other function. If the supports are non-overlapping, we have

$$\psi_{l',m'}((m-1)2^{-l}) = \psi_{l',m'}(m2^{-l}) = \psi_{l',m'}((m+1)2^{-l}) = 0. \tag{31}$$

Then the orthogonality is implied by Theorem B.2. In the second case, we assume that the support of $\psi_{l',m'}$ is contained within that of $\psi_{l,m}$ without loss of generality. In this case, it is not hard to see that we have either

$$[(m'-1)2^{-l'}, (m'+1)2^{-l'}] \subset [(m-1)2^{-l}, m2^{-l}]$$

or

$$[(m'-1)2^{-l'}, (m'+1)2^{-l'}] \subset [m2^{-l}, (m+1)2^{-l}].$$

In both situations, (31) is also true, which proves Statement 1.

Statement 2 is easy to prove by noting the boundary conditions $\psi_x(0) = \psi_x(1) = 0$ for all $x \in \mathbf{U} \setminus \{0, 1\}$ and the reproducing property. Statement 3 is a direct consequence of Statements 1 and 2, together with the matching dimensionality of the two linear spaces. $\qquad\square$

### B.3. Application to Gaussian processes with a Laplace correlation

Now consider a zero-mean Gaussian process with a correlation function

$$\text{Cov}(Z(x), Z(y)) = \exp\{-\theta|x - y|\} = \exp\{\theta(x \wedge y)\} \exp\{-\theta(x \vee y)\}.$$

Then the corresponding (28)-(29) are

$$A + Be^{-\theta(b-a)} + Ce^{-\theta(c-a)} = 0,$$
$$Ae^{-\theta(c-a)} + Be^{-\theta(c-b)} + C = 0,$$

which, together with the unit RKHS norm condition, determines a unique solution. Direct calculations show the solution

$$\phi_{a,b,c} = \begin{cases} 0, & x \le a, \\ \dfrac{2B\,\sinh\big(\theta(c-b)\big)}{\sinh\big(\theta(c-a)\big)}\,\sinh\big(\theta(x-a)\big), & a \le x \le b, \\ \dfrac{2B\,\sinh\big(\theta(b-a)\big)}{\sinh\big(\theta(c-a)\big)}\,\sinh\big(\theta(c-x)\big), & b \le x \le c, \\ 0, & x \ge c, \end{cases}$$

where

$$B = \left( \tfrac{1}{2}\Big[ \coth\big(\theta(b-a)\big) + \coth\big(\theta(c-b)\big) \Big] \right)^{1/2}.$$

Since we are only interested in the symmetric case with $c - b = b - a =: \Delta$, the above formula can be simplified as

$$\phi_{b-\Delta,b,b+\Delta} = \begin{cases} \sqrt{\dfrac{2}{\sinh(2\theta\Delta)}}\,\sinh\big(\theta(x-a)\big), & b - \Delta \le x \le b, \\ \sqrt{\dfrac{2}{\sinh(2\theta\Delta)}}\,\sinh\big(\theta(c-x)\big), & b \le x \le b + \Delta, \\ 0, & x \notin [b-\Delta, b+\Delta]. \end{cases}$$

This can be used to construct all the "wavelet-like" basis $\psi_x$. Specifically, they are

$$\psi_{lm} = \begin{cases} \sqrt{\dfrac{2}{\sinh(2^{1-l}\theta)}}\,\sinh\big(\theta(x-(m-1)2^{-l})\big), & (m-1)2^{-l} \le x \le m2^{-l}, \\ \sqrt{\dfrac{2}{\sinh(2^{1-l}\theta)}}\,\sinh\big(\theta((m+1)2^{-l}-x)\big), & m2^{-l} \le x \le (m+1)2^{-l}, \\ 0, & x \notin [(m-1)2^{-l}, (m+1)2^{-l}], \end{cases}$$

for $l = 1, 2, \ldots$, and $m = 1, 3, \ldots, 2^l - 1$.

In addition to $\psi_{lm}$, we need two basis functions to generate the "boundary space" $\text{span}\{K(\cdot, 0), K(\cdot, 1)\}$. We use a direct orthogonalization approach. We can take the symmetric choice, for $0 \le x \le 1$,

$$\psi_{01}(x) = \frac{\exp\{-\theta x\} + \exp\{-\theta(1-x)\}}{\sqrt{2(1 + \exp\{-\theta\})}},$$

$$\psi_{02}(x) = \frac{\exp\{-\theta x\} - \exp\{-\theta(1-x)\}}{\sqrt{2(1 - \exp\{-\theta\})}}.$$

Now all $\psi_{lm}$'s form an orthonormal basis of $\mathbf{V}$, and this basis is used in Theorem 1.

# C. Details of Implementation

In this section, we provide implementation details of SIKA-GP and the experimental setting.

**Bayesian fully-connected layer**: SIKA-GP consists of a Bayesian fully-connected (FC) layer, $y = \phi(x)\mathbf{w}$, and the sparsely activated basis $\phi(x)$. Following (Blundell et al., 2015), the random weight $\mathbf{w}$ is parameterized by its Gaussian mean and variance. To ensure a positive standard deviation, $\mathbf{w}$ is parameterized by $\{\mathbf{m}, \boldsymbol{\rho}\}$ as

$$\mathbf{w} = \mathbf{m} + \boldsymbol{\sigma} \odot \boldsymbol{\epsilon} = \mathbf{m} + \log\left(1 + \exp(\boldsymbol{\rho})\right) \odot \boldsymbol{\epsilon}, \ \boldsymbol{\epsilon} \sim \mathcal{N}(\mathbf{0}, \mathbf{I}), \tag{32}$$

where $\boldsymbol{\sigma} := \log\left(1 + \exp(\boldsymbol{\rho})\right)$ is the deviation and $\boldsymbol{\epsilon}$ is sampled independently from a standard normal distribution. To improve the sampling efficiency and reduce the variance of stochastic gradient descent, we incorporate Bayesian linear flipout (Wen et al., 2018) to sample $\boldsymbol{\epsilon}$ with the mini-batch data:

$$\mathbf{y} = \phi(\mathbf{X})[\mathbf{m} + \boldsymbol{\sigma} \odot \boldsymbol{\epsilon}] \tag{33}$$
$$= \phi(\mathbf{X})\mathbf{m} + (\phi(\mathbf{X}) \odot \mathbf{R})(\boldsymbol{\sigma}\boldsymbol{\epsilon}), \tag{34}$$

where $\mathbf{R}$ is the Rademacher random sign matrix ($\pm 1$ masks) multiplied by the mini-batch $\phi(\mathbf{X})$.

**Computing Infrastructure**: The software is built on `PyTorch`. The CPU benchmark experiments were run on M4 Pro with 48GB RAM, and the CUDA benchmark experiments were run on NVIDIA RTX4080.

## C.1. Time analysis

We benchmark the SIKA-GP with Sparse Indexing (SI) and compare it to the dense GP. For sparse SIKA-GP (ours), the sparsely activated basis functions are stored as a *sparse* tensor vector $\psi$ of shape `[B,D,L+2]`. For dense inference, the basis functions are stored as a *dense* tensor vector $\phi(x)$ of shape `[B,D,M]`, where $M = 2^L + 1$.

We record the inference time across varying dyadic levels $L$, batch sizes $B$, a range of MC samples $S$, and varying feature dimensions $D$ to demonstrate the improved efficiency and robustness of the sparse SIKA-GP. The results are shown in Figure 5 and 6.

## C.2. DGP on UCI regression

**Experiment Setup**    The DGP models are designed by stacking 2 hidden layers of SIKA-GP. The dyadic level of each hidden layer is set to $L = 7$, which corresponds to 129 inducing points. The data for the UCI regression are available at https://github.com/treforevans/uci_datasets. The model architecture and hyperparameters of the regression is shown in Table C.1.

*Table C.1.* Model architectures for regression on UCI datasets.

| Model | Hyper-parameter | Gas | Kin40K | Protein |
|---|---|---|---|---|
| DGP | Hidden layers | 2 | 2 | 2 |
| | Hidden features | 128 | 64 | 64 |
| | Dense basis | 129 | 129 | 129 |
| | Grid bounds | [0,1] | [0,1] | [0,1] |
| | Training epochs | 100 | 100 | 100 |
| | Batch size | 512 | 512 | 512 |
| | Training samples | 10 | 10 | 10 |
| | Testing samples | 20 | 20 | 20 |
| | Learning rate | 0.001 | 0.001 | 0.001 |
| | Weight decay | 0.0005 | 0.0005 | 0.0005 |
| DGP-SIKA | Hidden layers | 2 | 2 | 2 |
| | Hidden features | 128 | 64 | 64 |
| | Sparsely activated basis | 7 | 7 | 7 |
| | Grid bounds | [0,1] | [0,1] | [0,1] |
| | Training epochs | 100 | 100 | 100 |
| | Batch size | 512 | 512 | 512 |
| | Training samples | 10 | 10 | 10 |
| | Testing samples | 20 | 20 | 20 |
| | Learning rate | 0.001 | 0.001 | 0.001 |
| | Weight decay | 0.0005 | 0.0005 | 0.0005 |

**Metrics**    Consider a test dataset of size $T$, denoted by $\mathcal{D}_{\text{test}} = \{\mathbf{x}_t, y_t\}_{t=1}^{T}$, with $\hat{y}_t = \mathcal{M}(\mathbf{x}_t)$ as the prediction of model $\mathcal{M}$ given input $\mathbf{x}_t$. DGP models on regression tasks are evaluated using three criteria: Root Mean Squared Error (RMSE), Negative Log Predictive Density (NLPD), and the average runtime per epoch.

**RMSE** is used to evaluate predictive accuracy by quantifying the degree to which the predictions diverge from the actual target values.

$$\text{RMSE}(\mathcal{D}_{\text{test}}, \mathcal{M}) = \sqrt{\frac{1}{T} \sum_{t=1}^{T} (y_t - \mathbb{E}[\mathcal{M}(\mathbf{x}_t)])^2}, \tag{35}$$

where $\mathbb{E}[\mathcal{M}(\mathbf{x}_t)]$ is estimated by drawing $\tilde{s}$ samples from the learned variational distribution.

**NLPD** serves as a standard probabilistic measure for uncertainty quantification, which is the negative logarithm of the likelihood of the test data according to the predictive distribution. In terms of Gaussian noisy likelihood, NLPD is given by

$$\text{NLPD}(\mathcal{D}_{\text{test}}, \mathcal{M}) = -\sum_{t=1}^{T} \log p(y_t = \mu_t | \mathbf{x}_t) = \frac{1}{T} \sum_{t=1}^{T} \left[ \frac{(y_t - \mu_t)^2}{2\sigma_t^2} + \frac{1}{2} \log(2\pi\sigma_t^2) \right], \tag{36}$$

$$\mu_t = \mathbb{E}[\mathcal{M}(\mathbf{x}_t)], \ \sigma_t^2 = \mathbb{V}[\mathcal{M}(\mathbf{x}_t)]. \tag{37}$$

## C.3. DKL on image classification

**Experiment Setup** We use `PyTorch` (Paszke et al., 2019) to implement DNN feature extractors and `GPyTorch` (Gardner et al., 2018) to implement GP modules. In classification tasks, we apply a softmax likelihood to normalize the output digits into probability distributions. DNNs are non-Bayesian models trained via negative log-likelihood loss, while DKL and DAK models are trained via ELBO loss.

**MNIST**: The feature extractor is a simple CNN: `Conv2d(1,32,3)` $\rightarrow$ `Conv2d(32,64,3)` $\rightarrow$ `MaxPool2d(2)` $\rightarrow$ `Dropout(0.25)` $\rightarrow$ `Linear(9216,128)` $\rightarrow$ `Dropout(0.5)`. To make a fair comparison, for both SVDKL and DAK/DKL-SIKA, we applied the same number of inducing points, training epochs, and MC samples.

**CIFAR-10**: The feature extractor is a ResNet-18 followed by a linear embedding layer that compressed the 512 output features into 128 base GP channels.

**CIFAR-100**: The feature extractor is a ResNet-34 followed by a linear embedding layer to compress the 2048 output features into 512 base GP channels. For SVDKL, we use a pretrained ResNet-34 and fine-tune the last-layer GP since SVDKL struggled to fit using full training. For DAK and DKL-SIKA, we use both full-training and fine-tuning strategy.

The setting of all training tasks are described in Table C.2 and Table C.3. For DAK, we implemented DAK-MC using Monte Carlo estimation given the intractable softmax likelihood. We used dyadic inducing points with a level 7 as $\mathbf{U}^{[7]} = \{0, 1, 1/2, 1/4, 3/4, \ldots, 127/128\}$ of size 129 for each base GP component. For DKL-SIKA, the model has the same architecture and hyperparameters with DAK but with the inference method based on sparse indexing. For SV-DKL, we employed the `VariationalELBO` with `SoftmaxLikelihood` as the variational loss objective. `GridInterpolationVariationalStrategy` is applied within `IndependentMultitaskVariationalStrategy` to perform additive KISS-GP approximation. For each KISS-GP unit, we used 128 variational inducing points initialized on a grid of size $[0, 1]$.

**Metrics** The model's performance is evaluated through four standard metrics: Top-1 accuracy, ELBO, Negative Log Likelihood (NLL), and Expected Calibration Error (ECE).

**ECE** quantifies the degree of "calibration" of a probabilistic model in UQ, specifically for classification problems. It is defined as the weighted average of the absolute difference between the model's predicted probability (confidence) and the actual outcome (accuracy) over several bins of predicted probability. Mathematically, ECE is given by:

$$\text{ECE} = \sum_{m=1}^{M} \frac{|B_m|}{n} \left| \text{acc}(B_m) - \text{conf}(B_m) \right|, \tag{38}$$

where $M$ is the number of bins into which the confidence values are partitioned, $B_m$ is the set of indices of samples whose predicted confidence falls into the $m$-th bin, $n$ is the total number of samples.

## C.4. DKL on transformer-based language models

**Task and Dataset**: We evaluate intent classification and out-of-domain (OOD) detection on CLINC150. The dataset contains 150 in-domain (ID) intents and an additional out-of-scope class. In our processed version, ID labels correspond to intents $\{0, \ldots, 149\}$ and OOD examples are assigned label 150. We follow the standard split provided by the dataset (`train`/`validation`/`test`). For ID intent classification, we train only on ID examples by filtering out all instances with label 150 from the training split. For OOD detection, we form an evaluation set by combining the ID portion of the test split with the OOD portion of the test split.

**Input Preprocessing**: Each utterance is tokenized using the tokenizer associated with `distilbert-base-uncased`. We truncate or pad sequences to a fixed maximum length of $L_{\max} = 48$ tokens. Padding uses the tokenizer's default padding token and attention masks are provided to the encoder. All text is lowercased implicitly by the uncased tokenizer.

*Table C.2.* Model architectures for image classification on MNIST, CIFAR-10 and CIFAR-100.

| Model | Hyper-parameter | MNIST | CIFAR-10 | CIFAR-100 |
|---|---|---|---|---|
| SVDKL | Feature extractor | CNN | ResNet-18 | ResNet-34 |
| | NN out features | 64 | 128 | 128 |
| | Inducing points per feature | 128 | 128 | 128 |
| | Grid bounds | [0,1] | [0,1] | [0,1] |
| | Training epochs | 20 | 200 | 50 |
| | Training samples | 20 | 20 | 20 |
| | Testing samples | 100 | 100 | 100 |
| | Training strategy | Full-training | Full-training | Fine-tuning |
| DAK | Feature extractor | CNN | ResNet-18 | ResNet-34 |
| | NN out features | 64 | 128 | 512 |
| | Dense basis | 129 | 129 | 129 |
| | Grid bounds | [0,1] | [0,1] | [0,1] |
| | Training epochs | 20 | 200 | 200/50 |
| | Training samples | 20 | 20 | 20 |
| | Testing samples | 100 | 100 | 100 |
| | Training strategy | Full-training | Full-training | Full-training/Fine-tuning |
| DKL-SIKA | Feature extractor | CNN | ResNet-18 | ResNet-34 |
| | NN out features | 64 | 128 | 512 |
| | Sparsely activated basis | 7 | 7 | 7 |
| | Grid bounds | [0,1] | [0,1] | [0,1] |
| | Training epochs | 20 | 200 | 200/50 |
| | Training samples | 20 | 20 | 20 |
| | Testing samples | 100 | 100 | 100 |
| | Training strategy | Full-training | Full-training | Full-training/Fine-tuning |

*Table C.3.* Details of training optimizer for image classification on MNIST, CIFAR-10 and CIFAR-100.

| Optimization | MNIST | CIFAR-10 | CIFAR-100 |
|---|---|---|---|
| Optimizer | Adadelta | SGD | SGD |
| Initial lr. | 1.0 | 0.1 | 0.1 |
| Weight decay | 0.0001 | 0.0001 | 0.0001 |
| Scheduler | StepLR | CosineAnnealingLR | CosineAnnealingLR |
| Data Augmentation | MNIST | CIFAR-10 | CIFAR-100 |
| RandomCrop | - | size=32, padding=4 | size=32, padding=4 |
| HorizontalFlip | - | p=0.5 | p=0.5 |

**Backbone Encoder**: We use `distilbert-base-uncased` as the transformer encoder. Let $x$ denote an input utterance and let $\mathbf{H} \in \mathbb{R}^{L \times d}$ be the final-layer hidden states, with hidden size $d = 768$. We use CLS pooling to obtain a sentence representation

$$\mathbf{z} = \mathbf{H}_{1,:} \in \mathbb{R}^d, \tag{39}$$

where index 1 denotes the first token position (the CLS token). Following common practice for stable training and improved conditioning, we optionally apply a learned projection

$$\tilde{\mathbf{z}} = \mathbf{W}_p \mathbf{z} + \mathbf{b}_p, \qquad \mathbf{W}_p \in \mathbb{R}^{r \times d}, \tag{40}$$

with projection dimension $r = 256$ in our default setting.

**Predictive Head: DKL-SIKA (SIKA-GP on Transformer Features).** On top of transformer features, we attach a Gaussian process predictive layer implemented with SIKA. We treat the transformer as a feature map and define a GP over the projected representation $\tilde{\mathbf{z}}$. For multi-class intent classification with $C = 150$ ID intents, the head produces class logits whose predictive distribution is obtained via the GP posterior. Training uses a variational sparse GP objective (ELBO) with inducing variables, where SIKA parameterizes the kernel approximation to improve scalability in both training and inference while retaining GP-style uncertainty.

**Training Objective and Optimization**: We optimize a variational objective consisting of the classification likelihood and a KL regularizer

arising from the GP variational approximation. Concretely, for a minibatch $\mathcal{B}$, we minimize

$$\mathcal{L}(\mathcal{B}) = \mathcal{L}_{\mathrm{CE}}(\mathcal{B}) + \beta \cdot \frac{\mathrm{KL}}{N}, \tag{41}$$

where $\mathcal{L}_{\mathrm{CE}}$ is the cross-entropy over the $C = 150$ ID intents, KL is the variational KL term from the GP head, $N$ is the number of ID training examples, and $\beta$ controls the KL strength. We use KL warm-up by linearly increasing $\beta$ from 0 to 1 over the first 15% of training steps to stabilize early optimization.

We use AdamW for optimization with decoupled weight decay. Unless stated otherwise, we use a batch size of 32 and train for 20 epochs. We clip the global gradient norm to 1.0.

**Freezing schedule**: To reduce optimization instability when coupling a probabilistic head to a large encoder, we freeze the DistilBERT encoder for the first two epochs and train only the projection layer and GP head. Afterwards, we unfreeze the last two transformer layers and continue end-to-end fine-tuning with a smaller encoder learning rate. Concretely, we use learning rate $10^{-3}$ for the projection/head parameters and $2 \times 10^{-5}$ for the unfrozen encoder layers. We apply a linear learning-rate schedule with a short warm-up (6% of total steps).

**Inference and Uncertainty Scores**: At test time, we compute predictive probabilities for each example and derive uncertainty scores used for OOD detection. We report two standard uncertainty measures.

*Predictive entropy*. Let $\bar{\mathbf{p}} \in \mathbb{R}^C$ denote the predictive class probabilities. Predictive entropy is

$$H(\bar{\mathbf{p}}) = -\sum_{c=1}^{C} \bar{p}_c \log \bar{p}_c. \tag{42}$$

*Mutual information (epistemic proxy)*. We estimate epistemic uncertainty via mutual information between predictions and model parameters. With $T$ stochastic predictive draws (from the GP posterior predictive), let $\mathbf{p}^{(t)}$ be the probability vector for draw $t$ and $\bar{\mathbf{p}} = \frac{1}{T} \sum_t \mathbf{p}^{(t)}$. We compute

$$\mathrm{MI} = H(\bar{\mathbf{p}}) - \frac{1}{T} \sum_{t=1}^{T} H\left(\mathbf{p}^{(t)}\right). \tag{43}$$

Unless specified otherwise, we use $T = 20$ draws.

**Evaluation Metrics**: We report ID intent classification accuracy on the ID portion of the test set. For calibration, we compute negative log-likelihood (NLL) and expected calibration error (ECE) on ID and OOD subsets separately (as shown in Table 4). For OOD detection, we treat OOD as the positive class and use uncertainty scores as detection scores to report AUROC and AUPRC.

**Implementation Details and Reproducibility**: All experiments are implemented in PyTorch using the Hugging Face `transformers` and `datasets` libraries for model initialization and data loading. We fix random seeds for Python, NumPy, and PyTorch and report mean $\pm$ standard deviation over multiple runs (matching the main paper protocol). Unless otherwise noted, training speed (min/epoch) is measured as wall-clock time per epoch averaged over the full training run, excluding one-time data preprocessing and model download overhead.

# D. Additional Experiments

## D.1. 1D example

In this section, we design an example of 1-dimensional (1D) regression to illustrate the impact of MC sampling and the depth of the GP layers on the predictive distribution and uncertainty quantification.

**Data**: Training data are generated by a GP with zero mean and squared exponential (SE) kernel $k(x, x') = \exp(-(x - x')^2)$ in $[-3, 3]$. The Testing data are generated by the same GP but in a larger region of $[-5, 5]$. Therefore, the testing points $\mathbf{X}_{\mathrm{test}} \in [-3, 3]$ are in-distribution data, while $\mathbf{X}_{\mathrm{test}} \in [-5, -3) \cup (3, 5]$ are out-of-distribution (OOD) data.

**Model:** The DGP model consists of 2 hidden layers of SIKA-GP with a hidden feature dimension of 10. The DKL model consists of a feature extractor, which a MLP model with dimensions $1 \to 100 \to 50 \to 10$, and a last-layer SIKA-GP. The dyadic levels of all SIKA-GP layers are set to 6 (65 inducing points in total).

**Results**: Figure D.1 shows the predictive distribution of trained DGP and DKL models in the testing data. Utilizing a greater number of MC samples enhances the predictive distribution for both in-distribution and out-of-distribution testing data. However, in modern BNNs, it is common to choose only 1 MC sample due to the expensive computational cost, which may sacrifice uncertainty estimation. With accelerated inference of our SIKA-GP, it becomes practically feasible to estimate a more accurate posterior with more MC samples.

- Impact of MC samples: as the number of MC samples grows, both DGP and DKL models have less tendency to be over-confident. Empirically, with $S \geq 20$, the predictive posterior is reasonably well-behaved. Therefore, we chose $S = 20$ in experiments of image classification with DKL.

- Impact of GP depth: With identical MC samples, deeper GP layers result in an improved predictive posterior, particularly when $S$ is quite small. With an increase in $S$, the disparity in the predictive distribution between DGP and DKL decreases.

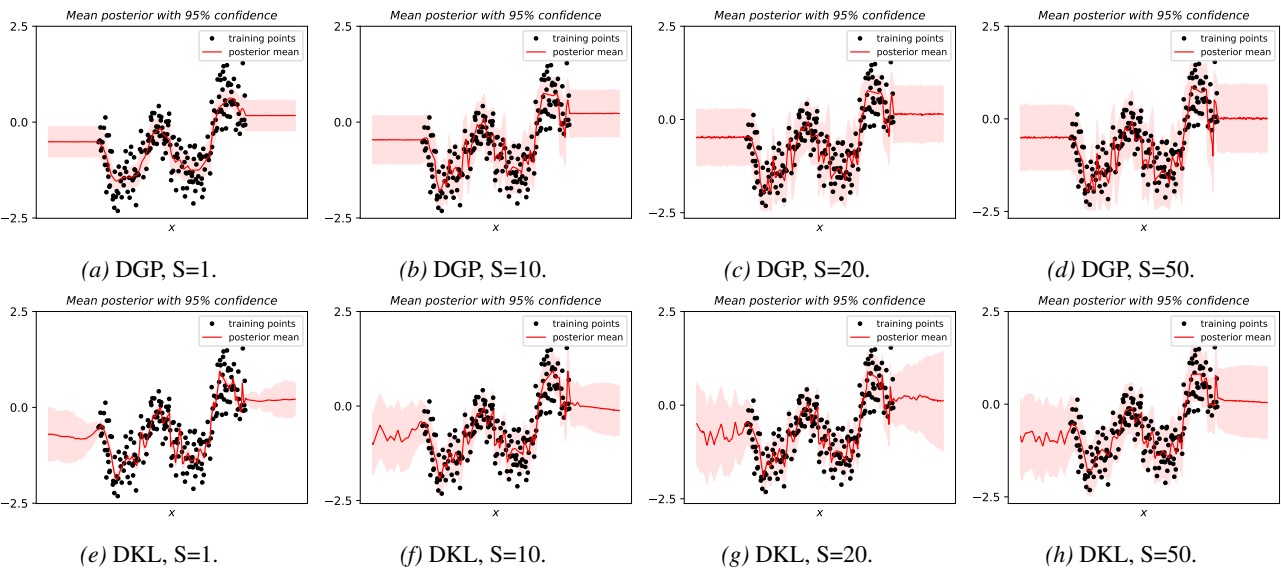

| *(a)* DGP, S=1. | *(b)* DGP, S=10. | *(c)* DGP, S=20. | *(d)* DGP, S=50. |
| *(e)* DKL, S=1. | *(f)* DKL, S=10. | *(g)* DKL, S=20. | *(h)* DKL, S=50. |

*Figure D.1.* 1D regression example of Deep SIKA-GP and DKL with different numbers of MC samples during training.

## D.2. Image classification

We provide additional experimental results on CIFAR-10/100 to show the effect of choosing the different numbers of MC samples during training.

*Table D.1.* Additional results of DKL-SIKA on image classification.

| Dataset | Hyper-parameter | Acc(%) | NLL | ECE(%) | Epoch time(s) |
|---|---|---|---|---|---|
| CIFAR-10 | $S = 1$ | 94.54 | 0.31 | 4.58 | **11.06** |
| | $S = 10$ | **95.13** | 0.47 | 4.50 | 13.85 |
| | $S = 20$ | 94.78 | **0.27** | **4.06** | 17.26 |
| CIFAR-100 | $S = 1$ | 75.69 | 1.75 | 6.52 | **21.22** |
| | $S = 10$ | 75.92 | 1.38 | 5.67 | 25.88 |
| | $S = 20$ | **76.94** | **1.25** | **4.31** | 36.31 |

**Model**: We use full-training on DKL-SIKA with different numbers of MC samples. The dimension of the extracted features is 128 and 512 for CIFAR-10 and CIFAR-100, respectively. The dyadic level of the SIKA-GP in the last layer is set to 7, which corresponds to 129 inducing points.

**Result**: While a higher number of MC samples may not directly result in increased accuracy, they do enhance uncertainty quantification, as indicated by the reduced NLL and ECE observed in Table D.1 when $S$ increases from 1 to 20. Thus, SIKA-GP provides an efficient method for drawing more MC samples in Bayesian deep learning without compromising the model performance.

