# OpenReview forum: "SIKA-GP: Accelerating Gaussian Process Inference with Sparse Inducing Kernel Approximations for Bayesian Deep Learning"
_ICML.cc/2026/Conference — ICML 2026 regular_

### Official Review · Reviewer_rxJL · 2026-02-23

**Soundness:** 3
**Presentation:** 3
**Significance:** 3
**Originality:** 3
**Overall Recommendation:** 4
**Confidence:** 2

**Summary:**

The manuscript proposes SIKA-GP, which promises to accelerate GP inference via BNN using sparse inducing kernel approximations based on a dyadic ordered template basis.

**Compliance With Llm Reviewing Policy:**

Affirmed.

**Final Justification:**

After careful consideration of my initial review and the authors' response, I recommend a "weak accept".

**Key Questions For Authors:**

The key motivation, "However, these approaches still face difficulties
when applied to very high-dimensional feature spaces or
when many inducing points are required, which limits both
computational efficiency and predictive accuracy." is unsatisfying. What are the fundamental advantages of the proposed algorithm?
While the experiments help, it is difficult to assess how the algorithm would perform compared to a relatively simple GP on a synthetic test function. Is the proposed algorithm universally better?

**Limitations:**

The authors included a "limitation" statement.

**Strengths And Weaknesses:**

Soundness:
Is the submission technically sound?
As far as I can assess, yes.


Are claims well supported?
Partly, for instance, “However, these approaches still face difficulties when applied to very high-dimensional feature spaces or when many inducing points are required, which limits both computational efficiency and predictive accuracy.” where “these approaches” refers to a limited number of prior similar methods. However, important prior work is left out, such as, large exact GPs (https://arxiv.org/abs/2411.05869), and the Vecchia approximation (https://www.jstor.org/stable/26997951).

Are the methods used appropriately?
Yes

If the paper includes theoretical results, are the proofs correct and based on reasonable assumptions?
Yes, as far as I can assess.

Are the experiments well-designed?
Partly, the results should, when applicable, have a well-designed base GP for comparison. When datasets are too large, the next best option (a pure inducing points method, for instance) can be used instead. It would be interesting to compare the method to a simple base GP on a relatively easy synthetic function to see where the actual improvement lies and whether it is universal.

Are the authors careful and honest about evaluating both the strengths and weaknesses of their work?
No, because of missing comparisons, the weaknesses of the algorithms might be hidden.
I suggest deeper comparative studies with some base methods to show the value of the algorithm.


Presentation:
Is the submission clearly written and well-structured?
Related work is insufficient.
The flow in section 4 is confusing. It is difficult to follow why and how the manuscript jumps from a simple stationary GP to BNN.
Prior work and novel contributions are also conflated (for example, in Variational Inference)


Is the overall narrative easy to follow?
No, see above

Does the work properly position itself in the context of prior/concurrent literature and clearly discuss how it differs?
The difference between other work in scalable GPs is difficult to follow.

Significance:
Does the paper address an important or relevant problem?
Yes, absolutely. Scalable GPs and BNNs have great value for ML.

Does it advance understanding, capabilities, or practice in machine learning?
Yes.

Could it influence future research or applications (e.g., other researchers or practitioners are likely to use the ideas or build on them)?
Yes.

Is the scope of impact broad or specialized, and is that appropriate for the contribution?
The impact scope is quite broad as BNNs are an important probabilistic learning tool.

Even if the improvements are modest or domain-specific, could they unlock new directions or provide practical utility?
Yes.


Originality:
Does the work provide new insights, deepen understanding, or highlight important properties of existing methods?
I can’t assess this because I am not very familiar with prior work in BNNs. It does little scale up pure GPs.

Does the work introduce new tasks, methods, theory, data, or perspectives that advance the field in some dimensions?
Yes.

Does this work offer a novel combination of existing techniques, and is the reasoning behind this combination well-articulated?
Yes.

Are the contributions clearly distinguished from closely related literature, and is the novelty well justified?
Difficult to assess in the manuscript's current form. A comparison to an exact base GP would be valuable, possibly on a new simpler test case if needed.

---

> ### Author Rebuttal · Authors · 2026-03-31
>
> We sincerely thank the reviewer for the constructive and thoughtful feedback. We appreciate your recognition of the contribution of our approach, as well as your insightful questions.
>
> > **[Q1]** "prior work on large exact GPs and Vecchia approximation"
>
> **[RQ1]** The authors thank the reviewer for pointing this out. Our
> intention was to refer specifically to prior inducing-point /
> sparse-basis GP *layers* that are most directly comparable to our
> setting, especially in large-scale Bayesian deep learning like DKL/DGP,
> rather than to the full scalable GP literature. We will revise the text
> accordingly and expand the related-work discussion to include
> complementary references.
>
> At the same time, our target setting is different. SIKA-GP is designed
> as a drop-in GP layer for deep Bayesian pipelines. As stated in (Risser et al.
> 2025), they calculate the variational variables in dense form in batches
> that are large enough to utilize the chosen hardware well. **Our
> contribution is therefore complementary: we accelerate this GP-layer
> regime by exploiting sparse dyadic basis activation and tensorized
> sparse indexing.** Moreover, we also give closed-form basis function to
> avoid dynamically solving inverse kernel matrix, which is not introduced
> in the previous work.
>
> > **[Q2]** "applicable base GP for comparison" and "synthetic function"
>
> **[RQ2]** The authors thank the reviewer for this suggestion. To our
> knowledge, a pure inducing point GP is not capable of learning features
> on image/language datasets, which is unstable and may even cause
> training failure (Sebastian W. Ober et al. 2021).
>
> Our experiments were designed to answer two focused questions: (i)
> whether SIKA delivers the claimed complexity reduction in the GP layer
> itself, and (ii) whether this translates into practical gains in modern
> DGP/DKL settings. **The scalability claim is not based only on fixed
> end-task settings, but is directly stress-tested in controlled runtime
> benchmarks.** For end-task comparisons, we tried our best to chose the
> most direct GP-layer baselines. In particular, DAK and DKL-SIKA are
> advanced GP baselines that can be used in deep Bayesian learning, and
> differ primarily in dense versus sparse GP execution. This isolates the
> effect of TSI rather than conflating it with other modeling changes. In
> this regime, the controlled scaling study over $(L,B,S,D)$ is a central
> piece of evidence rather than an auxiliary experiment.
>
> A base GP is more useful for comparison on simple synthetic functions,
> which has been tested in SVDKL and DAK. In this work, we focus on more
> efficient inference without any degradation from DAK. To better
> illustrate our proposed method, we will include a toy example on synthetic
> function with base GP in the revised version.
>
> **[Q3]** "weakness and suggest deeper comparative studies with some base methods"
>
> **[RQ3]** We thank the reviewer for raising this concern.
> - We would like to clarify that our intention was not to hide any
> weaknesses. We respectfully disagree that the current evaluation is not
> careful or honest; rather, it is targeted at isolating the specific
> computational contribution of SIKA-GP. Our **SIKA-GP is not a degradation
> from the existing variational GP methods, but improve the inference efficiency** by
> utilizing the sparse basis templates.
> - As mentioned in **[RQ2]**, we tried our best to chose the
> most advanced GP baselines. To the best of our knowledge, SVDKL (KISS-GP)
> and DAK are already the most advanced GP methods that are scalable
> enough for image classification and large language models. A simple base
> GP method could be unstable and cause training failure which has been
> studied before.
> - We explicitly discuss the limitations in Section 7. Lastly, we would appreciate if the reviewer could more
> specifically point out what weakness causes your concern and we would be happy to address that.
>
> > **[Q4]** "Related work and presentation"
>
> **[RQ4]** The authors thanks the suggestions and will revise accordingly.
> We will add discussion of complementary scalable GP baselines, and
> clarify that our target setting is different: we focus on GP layers that
> must be embedded into Bayesian deep models and executed through repeated
> stochastic forward passes on GPUs. So far, traditional GPs provide a
> mathematically beautiful approach to uncertain quantification but with
> limited impact in modern deep learning. We believe that one main reason
> is the lack of a route to readily apply various GP techniques in the
> mature BNN tool chain.
>
> Due to the page limitation, we put some derivations in the appendix. The derivation of dyadic kernel
> templates and orthonormal basis for generic Gauss-Markov kernels are
> introduced in **Appendix B.1-B.2**. The specific orthonormal basis functions
> for Laplace kernel is given in **Appendix B.3**. We will revise the main
> text and set more pointers in the revised paper to help readers better
> understand the theoretical derivations and necessary proofs.

---

> > ### Author Rebuttal · Reviewer_rxJL · 2026-03-31
> >
> > I am satisfied with the authors' answers and I will adjust my score accordingly.

---

> > > ### Author Response · Authors · 2026-04-01
> > >
> > > We appreciate your acknowledgement of our rebuttal and thank you for the constructive comments. We are pleased that our responses addressed your concerns.

---

### Official Review · Reviewer_rdKY · 2026-03-11

**Soundness:** 3
**Presentation:** 2
**Significance:** 3
**Originality:** 2
**Overall Recommendation:** 4
**Confidence:** 3

**Summary:**

This paper proposes SIKA-GP, a method for accelerating Gaussian Process inference using sparse inducing kernel approximations. The key idea is to construct a dyadic template basis that enables sparse kernel activations and efficient GPU computation. The method is also connected to Bayesian neural network formulations. Experiments show improvements in training and inference speed while maintaining competitive predictive performance.

**Compliance With Llm Reviewing Policy:**

Affirmed.

**Final Justification:**

Solved.

**Key Questions For Authors:**

1. How sensitive is the method to the number of inducing points or template basis elements?
2. Since the method uses a fixed dyadic grid rather than learned inducing locations, when does this design help or hurt?
3. If I understand correctly, the proposed method seems to be specific to the Laplace kernel. Could the authors comment on whether this limits its applicability in practice?

**Strengths And Weaknesses:**

Strengths:
- The paper tackles an important problem: improving the scalability of Gaussian Processes.
- The dyadic template kernel construction is interesting and seems to enable efficient sparse computation.
- The connection to BNN-style implementations is appealing.
- Experiments show consistent speed improvements with similar predictive performance.

Weaknesses:
- The theoretical analysis is limited; it would be helpful to better justify the approximation quality of the proposed kernel basis.
- Some parts of the method description are hard to follow, especially the construction of the dyadic kernel templates.
- Experiments could include stronger GP baselines or larger datasets to better demonstrate scalability.

---

> ### Author Rebuttal · Authors · 2026-03-31
>
> We sincerely thank the reviewer for the constructive and thoughtful feedback. We appreciate you find our work "important", "interesting", and "appealing".
>
> > **[W1]** "Theoretical analysis", "helpful to better justify the approximation quality"
>
> **[RW1]** The authors thank the reviewer for this insightful point.
> - **SIKA-GP is not a looser GP approximation** than the existing inducing-kernel model with the
> same inducing set. Let $V_U = \mathrm{span} (K(\cdot,u): u\in U)$. Our
> dyadic basis $\{\psi_{lm}\}$ is an orthonormal basis of $V_U$, so
> $K(x,U)K(U,U)^{-1}f(U)=\sum_{j=1}^M \psi_j(x) w_j$ is the same
> inducing-point approximation expressed in sparse coordinates.
> - **Computational gain**: compact support + dyadic ordering +
> tensorized sparse indexing (TSI) make only $L+2=O(\log M)$ basis
> functions active per input, so only the corresponding rows of $W$ need
> to be gathered and sampled.
> - **Bound on approximation gap**: $\| f(x) - \tilde{f}(x)\|_2=O(M^{-1})$, where $M=2^L+1$. We will add
> necessary discussions in the revision to support our proposed method.
>
> > **[W2]** "method description", "construction of the dyadic kernel templates ..."
>
> **[RW2]** The authors thank the reviewer for pointing this out. Due to page limits, the basis functions are derived and proved in **Appendix A**. The derivation of dyadic kernel templates and orthonormal basis for generic Gauss-Markov kernels are introduced in **Appendix B**.  We will set more pointers in the revised paper to help readers better understand the theoretical derivations and necessary proofs.
>
> > **[W3]** "include stronger GP baselines or larger datasets"
>
> **[RW3]** The authors thank the reviewer for this suggestion and agree
> that broader empirical validation would further strengthen the paper.
> - Our experiments were designed for two focused questions:
> (i) whether SIKA delivers the claimed complexity reduction in the GP
> layer itself, (ii) whether this translates into practical gains in
> modern DGP/DKL settings. The scalability claim is not based only on
> fixed end-task settings, but is directly stress-tested in controlled
> runtime benchmarks. For end-task comparisons, we tried our best to chose
> the most direct GP-layer baselines. This isolates the
> effect of TSI rather than conflating it with other modeling changes. In addition, we
> evaluate a transformer-based GP head on CLINC150 with a DistilBERT
> encoder, showing that the method applies beyond vision to
> high-dimensional learned representations.
>
> > **[Q1]** "sensitivity to the number of inducing points or template basis elements"
>
> **[RQ1]** The authors thank the reviewer for raising this concern.
> - In end-task experiments we fixed $M=129$ ($L=7$) because it provided
> stable predictive performance while keeping wall-clock comparisons
> practical across all tasks. The relative computational advantage of
> SIKA-GP should increase further as $M$ grows. If $M$ increases to 257, the training of SVDKL and DAK is much more slower, which trades too much efficiency for little performance. Additional ablation results with different dyadic levels in CIFAR-10 are shown below.
>
> | Metric | 1 | 2 | 4 | 7 | 10 |
> | --- | ---: | ---: | ---: | ---: | ---: |
> | ACC | 92.59 | 94.56 | 94.59 | 94.76 | 94.78 |
> | ECE | 5.79 | 4.17 | 4.78 | 4.10 | 4.06 |
> | NLL | 0.420 | 0.279 | 0.391 | 0.281 | 0.27 |
>
> > **[Q2]** Since the method uses a fixed dyadic grid rather than learned
> inducing locations, when does this design help or hurt?
>
> **[RQ2]** The authors thank the reviewer for raising this insightful
> point.
> - From an approximation accuracy point of view, there are two separate
> strategies to increase the accuracy. The first one is, as the reviewer
> mentioned, to learn the inducing locations. The second one, however, is
> to simply increase the number of inducing points, on a pre-specified
> finer grid. The second method is **much easier and has a
> theoretical guarantee**: as the inducing
> points become dense in the input region, the approximation will become
> exact.
> - However, the second approach would become difficult to use as the computational cost is
> usually problematic in high dimensions. The first approach can be
> viewed as a compromise in those situations, and that is why many
> existing methods chose to learn the inducing locations instead.
> - The major benefit of the fixed dyadic grid is that the **computation now
> scales at $O(\log M)$**, and a large number of inducing points can
> therefore be used efficiently.
>
> > **[Q3]** "whether Laplace kernel limits its applicability in practice"
>
> **[RQ3]** Although Laplace (also known as Matern 1/2) kernel is a restrictive
> kernel that loses certain capabilities (such as smoothness control), there is a significant gain in the model
> scalability. In practice, such limitation on applicability can be
> mitigated by DGP and DKL methods to learn more expressive features. For more task dependent constraints, high-order Matérn kernels can be extended based on basis templates.

---

> > ### Author Rebuttal · Reviewer_rdKY · 2026-04-02
> >
> > Solved and I will maintain my positive score.

---

> > > ### Author Response · Authors · 2026-04-02
> > >
> > > We appreciate your acknowledgement and constructive review. We are pleased that your concerns are solved.

---

### Official Review · Reviewer_d77c · 2026-03-11

**Soundness:** 3
**Presentation:** 3
**Significance:** 3
**Originality:** 3
**Overall Recommendation:** 5
**Confidence:** 4

**Summary:**

This paper proposes a framework called SIKA GP, which is designed to make GP uncertainty estimation more scalable and practical with GPUs. It leverages the sparse basis associated with the Laplace kernel to reduce the scaling with inducing points from O(M) to O(log M). Experiments are conducted on a selection of regression, vision and language benchmarks, with significant improvements in speed demonstrated for both training and inference.

**Compliance With Llm Reviewing Policy:**

Affirmed.

**Final Justification:**

Following the rebuttal, I have raised my score from 4 to 5, since the authors addressed my main concerns.

**Key Questions For Authors:**

Most experiments use a choice of M=129, would we perhaps see an even more compelling outcome if we illustrated larger values such as 257 in one or two cases?

What impact do we expect from limiting our choice of kernel, and is it quite task dependent?

**Limitations:**

Yes

**Strengths And Weaknesses:**

Strengths:

This paper tackles an important and challenging problem of scalability in Bayesian deep learning. It is clearly written and well presented. The experimental results are strong, and they span a suitably broad range of empirical tasks.

Weaknesses:

This technique leverages a property of the Laplace kernel, which is not applicable for RBF or higher order matern kernels. This is not necessarily problematic in itself, but the reader is left wondering how much performance we may have sacrificed. For example, how would SVDKL with RBF have fared across the experiments?  Also, as presently written, the abstract does not make this limitation of the method explicit, the claim to ‘accelerate GP inference’ comes across as more general.

---

> ### Author Rebuttal · Authors · 2026-03-31
>
> We sincerely thank the reviewer for the constructive and thoughtful feedback. We appreciate your recognition of the "importance", "well presentation" of our approach, the comprehensive experimental evaluation, and "strong experimental results".
>
> > **[W1]** "This technique leverages a property of the Laplace kernel,
> which is not applicable for RBF or higher order matern kernels. This is
> not necessarily problematic in itself, but the reader is left wondering
> how much performance we may have sacrificed. For example, how would
> SVDKL with RBF have fared across the experiments? Also, as presently
> written, the abstract does not make this limitation of the method
> explicit, the claim to ‘accelerate GP inference’ comes across as more
> general."
>
> **[RW1]** The authors thank the reviewer for raising concerns about the
> generality and rigor of our method.
>
> - **Applicability beyond Laplace kernel.** On one hand, we agree that
>   the Laplace (also known as Matern 1/2) kernel is a restrictive kernel
>   that loses certain capabilities of common GP kernels (such as
>   smoothness control). However, **on the other hand, there is a
>   significant gain in the model scalability**. So far, traditional GPs
>   provide a mathematically beautiful approach to uncertain
>   quantification but with limited impact in modern deep learning. We
>   believe that one main reason is the lack of a route to readily apply
>   various GP techniques in the mature BNN tool chain. Although SVDKL
>   with RBF provides combination of NN encoder and last-layer GP, the
>   training of SVDKL is still unstable and could even diverge, especially
>   when the dimension of features is very high. Our primary motivation is
>   to bridge this gap: to retain the Bayesian uncertainty and calibration
>   benefits of GPs while enabling them to scale to settings that require
>   large-scale NN or other flexible models.
>
> - **How much performance we may sacrifice.** To address the limitations
>   of Laplace (Markov) kernels on performance, we provide two
>   strategies. (1) **SIKA-GP can be stacked as in a deep Gaussian process
>   (DGP)**, which is used for the settings in Table 2; (2) **applying a NN encoder to extract the features before SIKA-GP** applied,
>   i.e., as in deep kernel learning (DKL) model, which is used for the
>   settings in Table 3, 4, and 5. It can be seen that SIKA-GP is indeed
>   effective in these settings. Both strategies enjoy much more efficient
>   training and inference, which can be challenging for other variational
>   GP models on large-scale and high-dimensional data. Our proposed
>   SIKA-GP significantly improves the computational efficiency without
>   sacrificing the uncertainty estimates and supports GPU pipelines,
>   which enjoy the same flexible design and computation benefits in BNNs.
>
> > **[Q1]** Most experiments use a choice of M=129, would we perhaps see
> an even more compelling outcome if we illustrated larger values such as
> 257 in one or two cases?
>
> **[RQ1]** The authors thank the reviewer for the helpful suggestion.
>
> In end-task experiments we fixed $M=129$ ($L=7$) because **it provided
> stable predictive performance while keeping wall-clock comparisons
> practical** across all tasks. Since only $L+2$ basis functions are active
> per input, sensitivity to $M$ is much milder than in other dense
> inducing-point methods, and the relative computational advantage of
> SIKA-GP should increase further as $M$ grows. If $M$ increases to 257,
> the training of SVDKL and DAK is much more slower, which trades too much
> efficiency for little performance. We provide additional ablation
> results on SIKA-GP with different dyadic levels in CIFAR-10 below.
>
> | Metric | 1 | 2 | 4 | 7 | 10 |
> | --- | ---: | ---: | ---: | ---: | ---: |
> | ACC | 92.59 | 94.56 | 94.59 | 94.76 | 94.78 |
> | ECE | 5.79 | 4.17 | 4.78 | 4.10 | 4.06 |
> | NLL | 0.420 | 0.279 | 0.391 | 0.281 | 0.27 |
>
> > **[Q2]** What impact do we expect from limiting our choice of kernel,
> and is it quite task dependent?
>
> **[RQ2]** The authors thank the reviewer for raising this point.
>
> As we mentioned in **[RW1]**, although Laplace (also known as Matern 1/2)
> kernel is a restrictive kernel that loses certain capabilities of some
> commonly used kernels (such as smoothness control), there is **a
> significant gain in the model scalability**. Besides, such limitation can
> be mitigated by DGP and DKL methods. For some specific task dependent
> constraints, we can extend to high-order Matern kernels based on basis
> functions of Laplace kernel.

---

> > ### Author Rebuttal · Reviewer_d77c · 2026-04-04
> >
> > Thank you for your thoughtful response. The added ablation over dyadic levels is helpful. Nonetheless, my primary concern about the modeling tradeoff is not fully resolved, nor is the framing of the abstract. The rebuttal argues that DGP/DKL mitigates the kernel restriction, and that SVDKL with RBF can be unstable in high-dimensional settings, but this is not quite the same as empirically demonstrating how much predictive performance is sacrificed relative to kernel choice. Even a limited comparison to an RBF on one or two representative tasks would make the case substantially stronger.

---

> > > ### Author Response · Authors · 2026-04-05
> > >
> > > Thank you for your acknowledgment of our rebuttal. For your follow-up question, we would like to make the following clarification:
> > >
> > > Empirical comparison with the RBF kernel is in fact contained in Table 3-5. We would like to point out that we included SVDKL as a standard DKL baseline, which serves as an empirical RBF-based reference in the deep-kernel setting. In our implementation, **SVDKL follows the official KISS-GP/SVDKL pipeline, and the GP layer uses the RBF kernel**. Thus, the current experiments indeed already provided an empirical comparison to an RBF on image and language tasks.
> > >
> > > Concretely, across the image benchmarks, **DKL-SIKA remains competitive with or slightly improves upon SVDKL in predictive performance while offering substantially lower training/inference cost**. We agree, however, that this is not a clean kernel-only ablation, and therefore provide an **additional ablation study on the kernel choice and inference acceleration below**:
> > >
> > > | Dataset   | Kernel Method    | ACC   | ECE  | NLL | Train Time | Infer Time |
> > > |-----------|------------------|-------|------|-----|------------|------------|
> > > | CIFAR-100 | RBF (dense)      | 77.43 | 5.36 | 0.98 | 37.38 | 9.78 |
> > > | CIFAR-100 | Laplace (dense)  | 76.13 | 4.18 | 0.99 | 35.62 | 9.66 |
> > > | CIFAR-100 | Laplace (SIKA)   | 76.61 | 3.45 | 0.94 | 20.08 | 7.07 |
> > >
> > > The main thesis of this work is regarding the **improvement in the computational efficiency**, therefore, the regression and classification performances alone under non-Laplace kernels without the speedup are not directly relevant to the main point of the work. The acceleration in the proposed strategy relies on some properties of the Laplace kernel (in Appendix A-B). We will therefore revise the paper to make this distinction explicit and avoid overclaiming.
> > >
> > > Our intended claim is not that Laplace (Matern 1/2) universally outperforms RBF, but that in deep-feature settings the predictive trade-off can be small while the computational gains are substantial. We are currently exploring the extension of the efficient computation pipeline to non-Laplace kernels, however, reaching log(M) complexity as well as the practical speedup are nontrivial for those kernels, which we had mentioned as a limitation of our current work.

---

### Official Review · Reviewer_mFfQ · 2026-03-12

**Soundness:** 3
**Presentation:** 3
**Significance:** 2
**Originality:** 3
**Overall Recommendation:** 4
**Confidence:** 3

**Summary:**

This work addresses the computational bottleneck of Gaussian Processes (GPs) in large-scale datasets by proposing a sparse inducing kernel approximation method based on dyadic ordered template basis functions. This work constructs compact and expressive kernel representations from sparsely activated bases, enabling efficient tensorized GPU computation and seamless integration with modern large-scale models and it is also embedded into BNNs with sparse activation to speedup in both training and infernece without sacrificing predictive performance. Empirical results on vision and transformer-based language benchmarks demonstrate that the approach has practical performance.

**Compliance With Llm Reviewing Policy:**

Affirmed.

**Key Questions For Authors:**

Look at the above "weaknesses" section.

**Limitations:**

yes

**Strengths And Weaknesses:**

Strengths:
1. This work addresses the computational bottleneck of Gaussian Processes (GPs) in large-scale datasets by proposing a sparse inducing kernel approximation method based on dyadic ordered template basis functions. The goal is to achieve O(logM) complexity inference while maintaining predictive performance, which is a practically relevant research direction.
2. This work proposes SIKA-GP, a compactly supported closed-form basis function-based method that accelerates Gaussian process inference, extends it to scalable deep feature learning for high-dimensional and large-scale datasets, and achieves significant speedups over existing GP baselines while preserving predictive accuracy.

Weaknesses:
1. The proposed idea of "sparse inducing kernel approximation + BNN equivalent representation" has been explored in previous works such as Ding et al. (2024) and Zhao et al. (2025). However, the paper fails to clearly define the essential differences between its core innovations and these works—only emphasizing "dyadic grid basis functions" and "O(logM) complexity" without explaining why existing methods cannot achieve this complexity or the irreplaceability of the dyadic grid design.

2. In understanding the technical details：1/ It is better to fully supplement the derivation process of basis function orthogonality in Appendix A, explicitly explaining how the Markov property of the Laplace kernel ensures the local support and orthogonality of basis functions.  2/ In limitation,only "limited expressiveness of the Laplace kernel" and "insufficient flexibility of fixed grids" are mentioned, without in-depth analysis of the specific impacts of these limitations in practical tasks (e.g., performance degradation on high-dimensional non-stationary data) or targeted mitigation strategies (e.g., adaptive kernel extension, hybrid grid design).

3. Regarding the experiment part：
1/ The experimental section only reports metrics such as accuracy and runtime without analyzing the quality of the method's uncertainty estimation (e.g., why SIKA-GP achieves a higher AUROC in OOD detection tasks).
2/ No ablation experiments are designed to verify the necessity of core modules (e.g., dyadic grid basis functions, TSI algorithm, Laplace kernel), making it impossible to prove the contribution of each module to performance and efficiency improvements.

4. Regarding the writing of the paper: 1/ Lack of Remark4.1, the numbering directly skipped from Theorem 4.1  to Remark4.2.
2/ Expressions such as "dyadic level", "level", and "inducing point level" alternate, lacking uniform standardization.

---

> ### Author Rebuttal · Authors · 2026-03-31
>
> We appreciate the reviewer’s constructive feedback and the recognition of our contribution. Below, we respond to the key concerns.
>
> **[W1]** "clearly define the essential differences between core innovations and previous work ..."
>
> **[RW1]** We thank the reviewer for raising this concern. We
> agree that our novelty needs to be stated more sharply.
> - **Prior work:** Ding et al. and Zhao et al. establish sparse
> expansions or inducing-kernel BNN equivalences, but **their forward
> computation pipeline remains dense** in the inducing dimension. The concept was firstly introduced in Ding et al, but it
> was largely a theoretical study where the computation pipeline was not
> investigated. Zhao et al. evaluated the computational cost in a deep kernel
> learning framework, but they still evaluate dense
> $\phi(x)\in\mathbb{R}^M$ or multiply/sample against the full width $M$.
> Their definitions heavily rely on the Cholesky decomposition of the
> inducing kernel matrix, which makes the closed-form sparse basis
> unachievable.
> - The underlying sparsity was also not sufficiently utilized, and there is a clear gap between the mathematical concept and
> real-world computation pipelines. So far, GPs provide a mathematically
> beautiful approach but with limited impact
> in modern DL. We believe that one main reason is the lack of
> an engineering route to readily apply GP techniques in the mature
> Bayesian DL tool chain.
> - **Our contribution is the missing computational bridge** that turns this
> structure into an efficient GP layer: (i) a dyadic ordered orthonormal
> basis with closed-form local support; (ii) the property of only one
> active basis per dyadic level; and (iii) TSI, which maps each input
> directly to the active indices. This is why we obtain $O(\log M)$
> *per-example forward/sampling cost*; a generic inducing set or arbitrary
> grid does not preserve the nested multiresolution structure required for
> closed-form sparse indexing. We plan to revise Sections 3–5 to separate
> clearly what is inherited from earlier induced-kernel formulations and
> what is new in SIKA-GP.
>
> **[W2]** "presenting technical details: better supplement for understanding technical details ..." and "limited Laplace kernel ..."
>
> **[RW2]** We thank the reviewer for these helpful suggestions.
> - **Derivation of basis function:** We prove the orthogonality of basis functions in Appendix A. The derivation of template basis function and orthonormal
> basis are introduced in Appendix B.1-B.2 in a principled framework. The
> specific orthonormal construction for the Laplace kernel is given in
> Appendix B.3, by specializing the general framework. We will help readers direct to Appendix A-B more explicitly in the revision.
> - **Kernel choice:** We agree that Laplace kernel (Matern 1/2) is not the most general
> kernel. However, this is _not_ a significant limiting factor. In DNNs, the
> ReLU activation is simple and restrictive, but it is widely used in most
> cases and even in advanced models. The Laplace kernel may
> lose certain capabilities in smoothness control, but there is a significant gain in the model scalability. Potential
> performance degradation can be mitigated in two
> ways: stack multiple layers of SIKA-GP as a deep GP or apply a NN
> encoder to learn expressive features as a DKL model. Both approaches are
> evaluated in Section 6, and the empirical results show that it does not
> sacrifice performance and indeed achieves scalability improvements.
> - **Future directions:** We thank the reviewer to point out some mitigation strategies
> (e.g., adaptive kernel extension, hybrid grid design). As mentioned
> in Section 7, they are promising future directions built on top of SIKA-GP, but we mainly focus on the improvement of
> computation in this work.
>
> **[W3]** "experiment: accuracy vs. uncertainty quality ..." and "ablation on core modules ..."
>
> **[RW3]** We thank the reviewer for raising this point.
> - **Uncertainty evaluation:** We not only report the accuracy and runtime, but also the
> commonly-used UQ metrics such as NLL, ECE. For AUROC, we use the same predictive entropy/MI scoring as the
> baselines, but a cheaper forward path allows a better-estimated
> posterior predictive.
> - **Ablation:** Firstly, we would like to clarify that some core components, including
> dyadic basis function, TSI, and Laplace kernel, are necessary
> and cannot be replaced to allow efficient tensor operations and take
> advantage of the ${O}(\log M)$ speedup. As for ablation, Fig 5-7 isolate the contribution
> of TSI by comparing dense vs. sparse forward execution over a wide range
> of $L,B,S,D$.
>
> **[W4]** "Writing inconsistency ..."
>
> **[RW4]** The authors thank the reviewer's detailed suggestion.
> - We follow the ICML-style templete and automatically number all
> theorems, propositions, and remarks using the same counter. If it causes
> significant confusion, we will revise to change the number schemes.
> - "Dyadic level $L$ corresponds to $2^L+1$ inducing points. We will
> provide the uniform terminology in the revised version.

---

> > ### Author Rebuttal · Reviewer_mFfQ · 2026-04-05
> >
> > I have reviewed the rebuttal. I will keep my previous rating.

---

### Decision · Program_Chairs · 2026-04-30

**Decision:**

Accept (regular)

**Comment:**

This paper presents an approximation to Deep Gaussian Processes (DGP) to enhance their scalability. In particular, this paper proposes sparse inducing kernel approximations with logarithmic scaling in the number of inducing points. On the one hand, the reviewers have a generally positive opinion of this paper. On the other hand, the reviewers point to some limitations in the positioning of the work, on experimental validation and limited applicability beyond Laplace kernels. The rebuttal clarified some of these issues and the reviewer kept a mildly positive opinion of the paper, except for one reviewer who recommends acceptance.

I share the reservations pointed by the reviewers. About the experimental validation, it would have been interesting to demonstrate the scalability with respect to the number of inducing points with experiments employing large M and reporting error vs wall-clock time/FLOPS. These are the type of plots that really show the quality of the approximation with respect to computational budget. In addition, experiments are limited to rather small datasets where it is difficult to appreciate the significant gains in performance/scalability characterizing the proposed method. Finally, it is important to report convincing experiments showing that the limitation of the framework being specific to the Laplace kernel does not represent a major limitation. I believe that these type of experiments should be included in the revision of this paper.